# Longitudinal multicompartment characterization of host-microbiota interactions in patients with acute respiratory failure

Georgios D. Kitsios [1,2] ✉, Khaled Sayed[3,4], Adam Fitch[2], Haopu Yang[5], Noel Britton[6], Faraaz Shah[1,7], William Bain [1,7], John W. Evankovich [1], Shulin Qin[1,2], Xiaohong Wang[1,2], Kelvin Li[2], Asha Patel[1,2], Yingze Zhang [1], Josiah Radder[1,2], Charles Dela Cruz[1], Daniel A. Okin [8], Ching-Ying Huang [8], Daria Van Tyne [9], Panayiotis V. Benos [3], Barbara Methé[1,2], Peggy Lai [8], Alison Morris[1,2,10] & Bryan J. McVerry [1,2,10]

Critical illness can significantly alter the composition and function of the human microbiome, but few studies have examined these changes over time. Here, we conduct a comprehensive analysis of the oral, lung, and gut microbiota in 479 mechanically ventilated patients (223 females, 256 males) with acute respiratory failure. We use advanced DNA sequencing technologies, including Illumina amplicon sequencing (utilizing 16S and ITS rRNA genes for bacteria and fungi, respectively, in all sample types) and Nanopore metagenomics for lung microbiota. Our results reveal a progressive dysbiosis in all three body compartments, characterized by a reduction in microbial diversity, a decrease in beneficial anaerobes, and an increase in pathogens. We find that clinical factors, such as chronic obstructive pulmonary disease, immunosuppression, and antibiotic exposure, are associated with specific patterns of dysbiosis. Interestingly, unsupervised clustering of lung microbiota diversity and composition by 16S independently predicted survival and performed better than traditional clinical and host-response predictors. These observations are validated in two separate cohorts of COVID-19 patients, highlighting the potential of lung microbiota as valuable prognostic biomarkers in critical care. Understanding these microbiome changes during critical illness points to new opportunities for microbiota-targeted precision medicine interventions.

Microbiota play a critical role in maintaining homeostasis and overall health. However, during critical illness, such as acute respiratory failure (ARF), microbial communities can be severely disrupted[1,2]. Such disruptions, characterized by deviations from a healthy microbial composition and diversity, may occur early in the hospital stay and have been associated with worse clinical outcomes[3–5]. Previous research has primarily focused on cross-sectional analyses of microbiota within individual body sites, neglecting potential interactions between different compartments and the longitudinal evolution of microbial communities. Moreover, the influence of patient-level factors and

therapeutic interventions, including antimicrobial therapies, on the microbiome of critically ill patients remains poorly understood, partly due to limitations of scale in studies published to date.

Conversely, precision medicine approaches in ARF have predominantly focused on host factors[6]. For instance, identifying distinct subphenotypes based on patterns of host response biomarkers measured in plasma samples (hyper- vs. hypo-inflammatory) has demonstrated prognostic value[7–9]. Hyperinflammatory patients exhibit elevated levels of injury and inflammation biomarkers, more severe organ dysfunction, worse prognosis, and may have differential responses to treatments[8]. However, the role of respiratory or intestinal microbiota in modulating host responses and their contributions to defined subphenotypes are still not well understood. Furthermore, limited data are available regarding the potential influence of respiratory microbiota on systemic host responses measured in plasma or localized inflammation within the lungs[10]. To advance precision medicine approaches that take into account the microbial side of the critically ill host, it is crucial to understand the dynamics of the microbiome and its relationship with host biological factors, clinical diagnoses, and therapeutic interventions in critical illness.

In this work, we conduct a longitudinal assessment of the microbiome in a large cohort of 479 ARF patients, specifically focusing on three key body sites: the oral cavity, lungs, and gut. By integrating bacterial and fungal community profiles with host response biomarkers measured in plasma and lower respiratory tract (LRT) samples, we demonstrate progressive dysbiosis in all three body sites and identify temporal associations between patient-level factors and therapeutic interventions on microbial communities. We further derive unsupervised clusters of microbiota and demonstrate significant associations with host responses and clinical outcomes, with lung microbiota clusters being the most predictive of survival. Finally, we validate our findings in two separate cohorts with a total of 146 patients with COVID-19-associated ARF.

## Results

### Cohort Description

We performed discovery analyses in a cohort of 479 patients with ARF who received invasive mechanical ventilation (IMV) via endotracheal intubation in UPMC Intensive Care Units (ICUs) (**UPMC-ARF cohort**), and then independent validation analyses in two cohorts of critically ill patients with COVID-19 pneumonia (49 patients at UPMC [**UPMC-COVID cohort**], and 97 patients at Massachusetts General Hospital ICUs, **MGH-COVID** cohort).

In the UPMC-ARF cohort, we enrolled patients with non-COVID etiologies of ARF between March 2015 and June 2022. We collected baseline research biospecimens within 72hrs from intubation, including blood for separation of plasma, oropharyngeal swabs (oral samples), endotracheal aspirates (ETA) collected for research or excess bronchoalveolar lavage fluid (BALF) from clinical bronchoscopy (lung samples), and rectal swabs or stool (gut samples)[3,11,12]. We repeated research biospecimen sampling between days 3–6 (middle interval) and days 7–12 (late interval) post-enrollment for subjects who remained in the ICU. We extracted DNA and performed next-generation sequencing (bacterial 16S rRNA gene sequencing [16S-Seq] for all available samples; fungal Internal Transcribed Spacer sequencing [ITS-Seq] targeting the regions 1 and 2 of the ITS rRNA gene, and Nanopore DNA metagenomics for a subset of samples) to profile microbiota in the oral, lung and gut communities, respectively[3,12,13]. We measured biomarker proteins in plasma samples and a subset of ETA/BALF supernatants with Luminex panels to profile systemic and regional (lung) host responses[7,10].

Patients had a median (interquartile range) age of 59.6 (46.7–68.7) years, 54.4% were men and 90.2% were white (Table 1). At the time of enrollment, 25.0% of patients were diagnosed with Acute Respiratory Distress Syndrome (ARDS per the Berlin definition[14]) 39.8% with pneumonia, 86.8% were receiving systemic antibiotics, and 64.8%

received corticosteroids for various indications. By 60 days, 26.9% of patients had died. Among the 350 patients who survived hospitalization, 48.8% were discharged to their home, with the remainder requiring additional longer-term care.

In the UPMC-COVID cohort from April 2020 through February 2022, we enrolled 49 patients with COVID-19 ARDS requiring IMV and obtained longitudinal plasma and ETA samples at baseline, middle, and late intervals (Table S1). We performed 16S sequencing for bacteria and measured host response biomarkers in both sample types. In the MGH-COVID cohort from April 2020 to May 2021, we enrolled 97 hospitalized patients, obtained serial lung (sputum or ETA) and stool (gut) samples (Table S1), and performed Illumina metagenomics[15]. To contextualize microbiota analyses from critically ill patients, we incorporated previously generated 16S-Seq data from upper respiratory tract (URT), LRT, and stool samples collected from healthy volunteers (**Healthy Controls**) that had been analyzed in smaller cross-sectional studies from our group[11,12].

### Progressive dysbiosis of microbial communities in three body compartments

By Illumina MiSeq 16S-Seq, we analyzed a total of 2557 clinical samples in the UPMC-ARF and UPMC-COVID cohorts and healthy controls, as well as 233 experimental control samples obtained either during patient sampling at the bedside or during sample processing in the laboratory. In an initial quality control step, we demonstrated robust detection of bacterial 16S reads in oral, lung, and gut samples in the UPMC-ARF cohort compared to negative controls (Figure S1). We considered clinical samples that generated ≥1000 quality 16S-Seq reads and performed rarefaction at 1000 reads to control for uneven sequencing depth between samples in the estimation of diversity indices[16–18] alpha diversity by Shannon index (Figure S1) and beta diversity by Bray-Curtis similarity index in centered-log ratio (CLR) transformed abundances. We also found that rectal swabs not coated by stool ("unsoiled" swabs) had systematic differences in bacterial load (16S rRNA gene copies by qPCR) and beta diversity compared to stool or visibly "soiled" rectal swabs (Figure S1). Therefore, we excluded "unsoiled" rectal swabs from further analyses because they may not offer sufficient representation of gut microbiota[11].

Following these quality-control steps, we first performed intra-compartment comparisons of samples from critically ill patients from the UPMC-ARF cohort to healthy control samples. At baseline, critically-ill patients had significantly lower alpha diversity in each compartment compared to corresponding healthy control samples. Despite the low Shannon index at baseline for ICU patients, their Shannon index further declined in all three body compartments in longitudinal samples (Fig. 1A). Similarly, baseline ICU samples had markedly significant differences in beta diversity from healthy controls (Fig. 1B). By taxonomic comparisons of CLR-transformed abundances within each compartment at baseline, ICU patient samples showed depletion of multiple commensal taxa, with significant enrichment for *Staphylococcus* in oral and lung samples, and *Anaerococcus* and *Staphylococcus* in gut samples (Fig. 1C).

We then performed inter-compartment comparisons among ICU samples. Bacterial load quantification by 1 S qPCR confirmed that the LRT (lungs) had significantly lower biomass compared to URT (oral) and gastrointestinal tract (stool or soiled rectal swabs, Fig. 1D). By beta-diversity comparisons (Bray-Curtis indices), oral and lung communities had high compositional similarity, whereas gut samples were compositionally different compared to oral and lung microbiota (Fig. 1E). Taxonomic comparisons of CLR-transformed abundances between compartments revealed that no specific taxa were systematically different between oral and lung microbiota (Fig. 1F), whereas in gut-lung comparisons, lung communities were enriched for typical respiratory commensals (e.g. *Rothia*, *Veillonella*, *Streptococcus*) and gut communities for gut commensals (e.g. *Bacteroides*,

**Table 1 | Baseline characteristics of enrolled mechanically ventilated patients in the UPMC-ARF cohort, stratified by 60-day mortality**

|  | All | Survivors | Non-Survivors | p |
|---|---|---|---|---|
| N | 479 | 350 | 129 |  |
| Age, years (median [IQR]) | 59.6 [46.7, 68.7] | 57.1 [44.1, 67.1] | 65.3 [55.8, 72.2] | **<0.01** |
| Men, n (%) | 256 (54.4) | 180 (52.6) | 76 (58.9) | 0.26 |
| Whites, n (%) | 425 (90.2) | 307 (89.8) | 118 (91.5) | 0.67 |
| BMI (median [IQR]) | 29.4 [25.5, 36.0] | 29.6 [25.5, 35.7] | 28.6 [25.3, 36.6] | 0.98 |
| COPD, n (%) | 104 (22.1) | 75 (21.9) | 29 (22.5) | 1.00 |
| Diabetes, n (%) | 168 (35.7) | 122 (35.7) | 46 (35.7) | 1.00 |
| Alcohol use, n (%) | 84 (17.9) | 60 (17.5) | 24 (18.9) | 0.84 |
| Immunosuppression, n (%) | 105 (22.3) | 71 (20.8) | 34 (26.4) | 0.24 |
| ARDS, n (%) | 117 (25.2) | 81 (24.0) | 36 (28.1) | 0.23 |
| WBC (median [IQR]) | 12.0 [8.7, 16.8] | 11.4 [8.1, 15.8] | 14.4 [10.1, 18.7] | **<0.01** |
| Creatinine (median [IQR]) | 1.2 [0.8, 2.3] | 1.1 [0.8, 2.0] | 1.6 [0.9, 2.5] | **0.01** |
| Plateau Pressure (median [IQR]) | 20.0 [16.0, 25.0] | 19.0 [16.0, 24.0] | 22.0 [18.0, 27.0] | **<0.01** |
| PaO2:FiO2 ratio (median [IQR]) | 164.0 [117.0, 206.0] | 168.0 [121.5, 211.0] | 157.0 [108.0, 205.0] | **0.04** |
| SOFA scores (median [IQR]) | 6.0 [4.0, 9.0] | 6.0 [4.0, 8.0] | 8.0 [5.0, 10.0] | **<0.01** |
| LIPS score (median [IQR]) | 5.5 [4.0, 6.5] | 5.0 [4.0, 6.5] | 6.0 [5.0, 7.5] | **<0.01** |
| Hypoinflammatory subphenotype, n (%) | 344 (75.6) | 254 (77.4) | 90 (70.9) | 0.18 |
| VFD (median [IQR]) | 22.0 [13.0, 25.0] | 23.0 [20.0, 25.2] | 0.0 [0.0, 19.0] | **<0.01** |

We compared continuous variables with non-parametric Wilcoxon tests and categorical variables with Fisher's exact tests between the three groups. Statistically significant differences (p < 0.05) are highlighted in bold. Source data are provided as a Source Data file.

IQR Interquartile Range, BMI body mass index, COPD chronic obstructive pulmonary disease, LIPS lung injury prediction score, WBC white blood cell count, PaO2 partial pressure of arterial oxygen, FiO2 Fractional inhaled concentration of oxygen, SOFA sequential organ failure assessment, VFD ventilator free days, ARDS acute respiratory distress syndrome. Immunosuppression was broadly defined as receipt of chronic steroids, alkylating agents, antimetabolites, biologics, calcineurin inhibitors, mycophenolate, active chemotherapy for cancer, or diagnosis of untreated immunodeficiency.

*Lachnoclostridium, Lachnospiraceae_uncl*) (Fig. 1F). In a limited comparison of two subjects with available synchronous ETA and BAL samples, high compositional concordance was shown for one subject in whom LRT community dominance by *Achromobacter* was shown for both ETA and BAL analysis, whereas for the other subject, taxonomic overlap between ETA and BAL was more limited (Figure S2).

Motivated by prior research supporting the enrichment of the lungs with gut-origin bacteria in ARDS[19], as well as emerging evidence for the role of a gut-lung axis in critical illness[20], we specifically tested whether certain patients had enrichment for gut-origin bacteria in their oral or lung samples, despite the global dissimilarity of the lung and gut compartments. We found that 4.5% and 8.3% of oral and lung samples, respectively, had >30% relative abundance for gut-origin bacteria (Fisher's test p = 0.03, Figure S3A), with progressive enrichment over time (Fisher's test p = 0.02, Figure S3B) in lung samples. Importantly, the gut-origin taxa enrichment in these lung samples could not be fully explained by oropharyngeal colonization with such taxa (Figure S3C). Taken together, these multi-site analyses point to the oral cavity as the primary source of lung microbiota, which could be seeded by micro-aspiration along the respiratory tract's gravitational gradient. At the same time, our analyses also provided evidence for gut-origin bacteria enrichment in the LRT in a subset of critically ill patients.

We then sought to understand the longitudinal composition of microbial communities when classified into clinically relevant categories of bacterial taxa. Recent evidence implicates loss of commensal anaerobic bacteria from the gastrointestinal or respiratory tract with adverse outcomes in critical illness[12,21,22]. Therefore, we classified bacteria in terms of their oxygen requirements (obligate anaerobes, facultative anaerobes, aerobes, microaerophiles, variable or unclassifiable, details in Table S2). Additionally, we classified bacteria by plausible respiratory pathogenicity (oral commensals, recognized respiratory pathogens, or other) due to their direct implications in prevalent or incident pneumonia in the ICU (Table S3)[12,23,24]. In both

oral and lung communities, we found a progressive decline in the CLR-transformed abundance of obligate anaerobes over time. There was, however, no corresponding change in the gut composition of anaerobic (obligate or facultative) bacteria over time (Fig. 2A, B). Stratified by plausible pathogenicity, we found a progressive decline of oral commensal bacteria in all three compartments, with a corresponding increase in pathogen abundance (Fig. 2C, D). The top representative taxa in each compartment are shown in Figure S4.

Beyond 16S-Seq, Illumina MiSeq Fungal ITS-Seq showed that >50% of communities in all three compartments were dominated by *C. albicans* (defined as >50% relative abundance), with a progressive decline in fungal Shannon index in oral and lung communities during follow-up (Figure S5). Nanopore DNA metagenomics of lung samples provided similar bacterial representations to 16S-Seq analyses and confirmed a high abundance of *C.albicans* detected by ITS-Seq (Figure S5). Thus, our analyses revealed a pattern of compartment-wide dysbiosis in ICU patients, with a progressive decline in diversity and enrichment for plausible pathogenic bacteria and *C.albicans*. We then sought to understand whether patient-level variables accounted for baseline or longitudinal dysbiosis.

**Clinical diagnoses and antibiotic exposure correlate with microbial community diversity and composition**

We constructed linear regression models with ecological metrics indicative of dysbiosis as outcomes (baseline Shannon index, bacterial load by 16S qPCR, obligate anaerobe and respiratory pathogen abundance) and clinical variables as predictors (Figure S6). History of COPD, immunosuppression, and clinical diagnosis of pneumonia showed the most significant associations with dysbiosis features, e.g. lower Shannon and anaerobe abundance in oral and lung communities for patients with COPD, and increased pathogen abundance in oral and gut communities for patients with a history of immunosuppression (Figure S6). History of immunosuppression was also associated with a higher abundance of *C. albicans* in oral and lung samples (Figure S5).

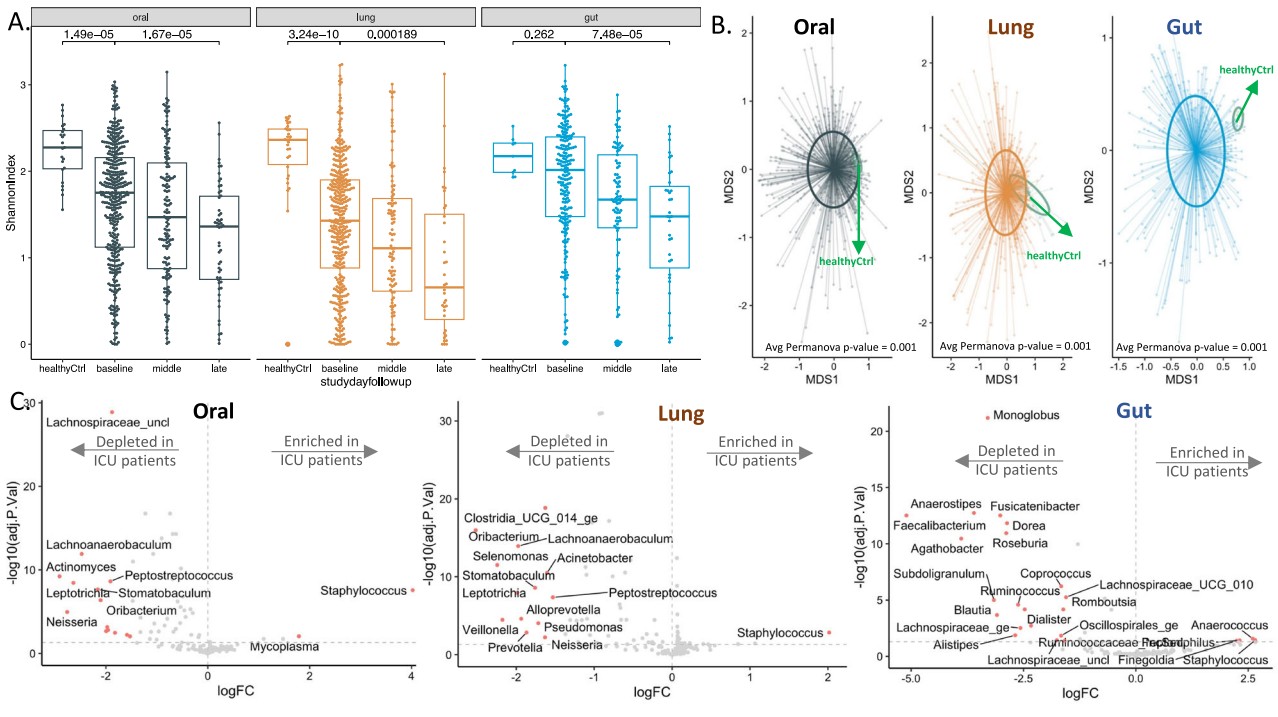

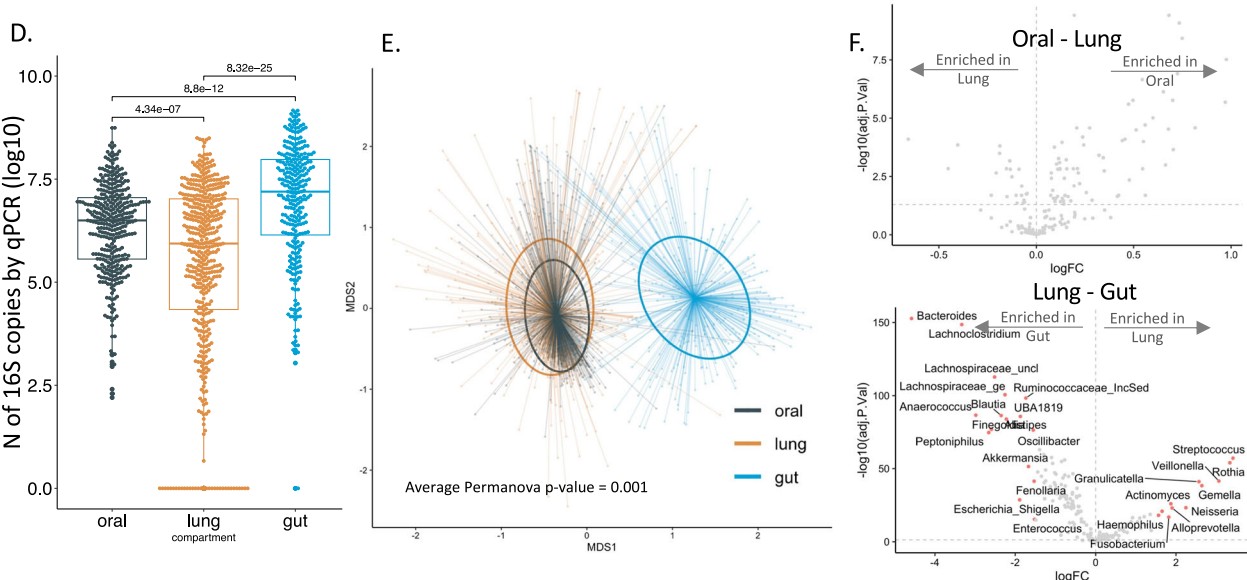

To further explore iatrogenic forces on microbiota composition, we focused on two common treatments in the ICU: antibiotics and steroids. We assessed antibiotic usage by i) anaerobic coverage, ii) a numerical scale that included duration, timing, and type (convex antibiotic score)[25], and iii) the Narrow Antibiotic Treatment (NAT) score[12,26]. We quantified steroid use as the daily equivalent dosage of prednisone in milligrams. Antibiotic usage was associated with bacterial burden in oral samples, as well as anaerobe and pathogen abundance in baseline gut samples (Figure S6). To explore the effects of antibiotics and steroids over time, we employed mixed linear regression models using longitudinal samples. In all three compartments, the receipt of anaerobic spectrum antibiotics was associated with a progressive decrease in obligate anaerobe abundance, as well as increased abundance of pathogens in the gut compartment (Table S4). Notably, antibiotic exposure quantified by the NAT and the convex antibiotic score was also significantly associated with a reduction in

anaerobe abundance and an increase in pathogen abundance within the gut microbiota.

## Microbial communities in each compartment form distinct clusters of diversity and composition

We next examined the microbial communities independent of clinical variables to capture important features directly from microbiome data. Within each compartment, we used Dirichlet Multinomial Mixture (DMM) models for 16S-Seq data ("bacterial DMM clusters") and defined that a three-class model offered optimal classification in each compartment, with striking differences in alpha diversity and composition between clusters (Fig. 3A). Cluster 1 in each compartment had high Shannon index in the range of healthy controls (referred to as High-Diversity cluster), cluster 3 had low Shannon index (Low-Diversity cluster), and cluster 2 had intermediate diversity (Intermediate-Diversity cluster). Low-diversity clusters had a markedly higher

**Fig. 1 | Intra- and inter-compartment comparisons of microbiota profiles by Illumina 16S-Seq reveal features of dysbiosis in all three body compartments in critically ill patients. Panels A–C:** Intra-compartment comparisons between ICU patients and healthy controls. **A** Samples from critically ill patients had significantly lower alpha diversity (Shannon index obtained post-rarefaction with random subsampling of reads in samples with ≥1000 16S rRNA gene reads) compared to corresponding healthy control samples in each compartment (Wilcoxon test $p < 0.001$), with a further decline of Shannon index over time in longitudinal samples in critically ill patients (Wilcoxon test $p < 0.001$). **B** Baseline samples from critically ill patients had markedly significant differences in beta diversity (Bray-Curtis indices in centered-log ratio transformed [CLR] abundances following random subsampling of reads in samples with ≥ 1000 reads) compared to healthy controls (visualized with Principal Coordinates Analysis [PCoA] and statistically compared with permutational analysis of variance [permanova] $p$ values < 0.001, adjusted for multiple comparisons with the Bonferroni method). **C** Taxonomic composition comparisons with the *limma* package showed high effect sizes and significance thresholds (threshold of log2-fold-change [logFC] of CLR-transformed abundances >1.5; Benjamini-Hochberg adjusted $p$ value < 0.05), revealing depletion for multiple commensal taxa in critically ill patients samples, with significant enrichment for *Staphylococcus* in oral and lung samples, and *Anaerococcus* and *Staphylococcus* in gut samples (significant taxa shown in red in the volcano plots). **Panels D–F:** Inter-compartment comparisons among ICU patients. **D** Lung samples had lower bacterial burden compared to oral and gut samples by 16S rRNA gene qPCR (all Wilcoxon test $p < 0.001$). **E** PCoA plot of beta-diversity shows compositional similarity for the oral and lung compartments, which were compositionally dissimilar to gut samples (permanova $p < 0.001$). **F** Taxonomic comparisons between compartments revealed that no specific taxa were systematically different between oral and lung microbiota above the threshold of logFC≥1.5, whereas in gut-lung comparisons, lung communities were enriched for typical respiratory commensals (e.g. *Rothia, Veillonella, Streptococcus*) and gut communities for gut commensals (e.g. *Bacteroides, Lachnoclostridium, Lachnospiraceae*). Source data are provided as a Source Data file. Displayed data include 583 oral, 543 lung, and 343 gut samples from ICU patients, as well as 23 oral, 32 lung, and 7 gut samples from healthy controls. Data displayed as boxplots with individual dots have their median as the line inside the box, interquartile range (25th–75th percentile) as the box itself, whiskers extend to 1.5 times the interquartile range, and individual dots beyond whiskers signify outlier observations. All statistical tests were two-sided.

abundance of pathogens and a lower abundance of anaerobes (Fig. 3B, C). In cross-compartment comparisons, DMM cluster membership was strongly associated between oral and lung communities (odds ratio of membership in the Low-Diversity cluster in both compartments 7.67, 95% confidence interval [4.22–14.25], $p < 0.0001$), whereas lung and gut clusters were less strongly associated although statistically significant ($p = 0.04$, Fig. 3D). Representative taxa in each cluster are shown in Figure S7. In longitudinal analyses, cluster membership showed relative stability for all compartments, with most samples assigned to the Low-Diversity cluster at baseline being assigned to Low-Diversity in the middle interval as well (75% of oral, 79% of lung, and 84% of gut samples, respectively, Figure S8). We next examined how these microbial communities related to host responses and clinical outcomes.

### Lung microbiota correlate with systemic host responses

We examined host-microbiota interactions with two independent approaches, a microbiota- and a host-centric approach. In the microbiota-centric approach, we correlated the top 20 abundant taxa in each compartment with systemic (plasma) and lung-specific (ETA/BALF supernatants) host response biomarkers. We found several significant correlations (Figure S9A–C), with typical pathogens correlating with ETA or plasma inflammatory biomarkers, such *Klebsiella* or *Staphylococcus* genera positively correlating with ETA fractalkine and Ang-2 levels, whereas *Escherichia-Shigella* abundance correlating with plasma TNFR1 and IL-6 levels. Conversely, typical oral commensals (e.g. *Rothia, Streptococcus, Prevotella* etc.) were inversely correlated with plasma sTNFR1 or sRAGE. In cluster comparisons, the bacterial DMM Low-Diversity cluster in the lungs was significantly associated with higher plasma sTNFR1, sRAGE and procalcitonin levels (Figure S9D).

In the host-centric approach, we applied a widely validated framework of host-response subphenotypes based on plasma biomarkers[7,27]. With a 4-biomarker parsimonious model (using sTNFR1, Ang2, procalcitonin and bicarbonate levels)[27], we classified individuals at baseline into a hyperinflammatory (22.9%) vs. a hypoinflammatory (77.1%) subphenotype. We found no significant relationship between host subphenotypes and DMM microbiota clusters in any compartment (Figure S10), but hyperinflammatory patients had higher pathogen abundance in lung communities ($p = 0.02$). To further investigate this association, we stratified patients by pneumonia diagnosis. We discovered that hyperinflammatory patients without pneumonia had higher pathogen abundance in lung samples compared to hypo-inflammatory patients ($p = 0.008$, Figure S10). These notable associations between lung pathogen abundance and the hyperinflammatory subphenotype imply that systemic subphenotypes might stem, at least in part, from undiagnosed pneumonia or respiratory dysbiosis.

### Lung microbiota clusters predict survival independent of clinical variables and host responses

Comparisons of microbial communities between survivors and non-survivors at 60-days post-ICU admission showed highly significant differences in Shannon index, obligate anaerobe and pathogen abundance in lung samples both at baseline and in the middle follow-up interval ($p < 0.05$, Figure S11). Additionally, lung samples with gut-origin taxa enrichment at baseline (defined as >30% relative abundance) showed markedly worse survival ($p = 0.004$, Figure S3). Survivors had a higher abundance of anaerobes and a lower abundance of pathogens in baseline oral samples (all $p < 0.0.5$, Figure S11), but no differences in gut profiles. Oral and lung communities of survivors were also found to have a lower abundance of *C.albicans* by ITS-Seq (Figure S5).

Beyond these cross-sectional comparisons of dysbiosis features, we sought to understand the impact of longitudinal changes on patient survival. To that end, we employed joint modeling[28], a powerful approach that combines longitudinal and survival analysis models. We examined longitudinal quantitative exposures of dysbiosis features (Shannon index, bacterial load, anaerobe, and pathogen abundance - details in Methods) in mixed linear regression models in each compartment, and then assessed the impact of longitudinal changes on 60-day survival. Mixed linear regression models demonstrated a progressive decline of the Shannon index in all three compartments, with progressive depletion of anaerobes and enrichment for pathogens in the oral and lung compartments (Fig. 4). By Cox proportional hazards models adjusted for age, baseline Shannon index and anaerobe or pathogen abundance in the oral and lung compartments were significantly associated with 60-day survival. Integration of longitudinal and survival analyses with joint modeling showed borderline significant effects for pathogen abundance in the oral compartment and anaerobe abundance in the lung compartment. Nonetheless, baseline microbiota features had stronger effect sizes associated with survival.

Apart from individual features, we also examined the impact of the global classification of microbial profiles by bacterial DMM clustering. In both oral and lung compartments, the Low-Diversity clusters were associated with worse 60-day survival in Kaplan-Meier curve analyses, whereas gut clusters had no survival impact (Fig. 4A–C). Notably, the prognostic effects of the Low-Diversity bacterial DMM cluster in the lungs remained significant after adjustment for age, sex, history of COPD, immunosuppression, severity of illness by SOFA scores, and host-response subphenotypes (adjusted Hazards Ratio-

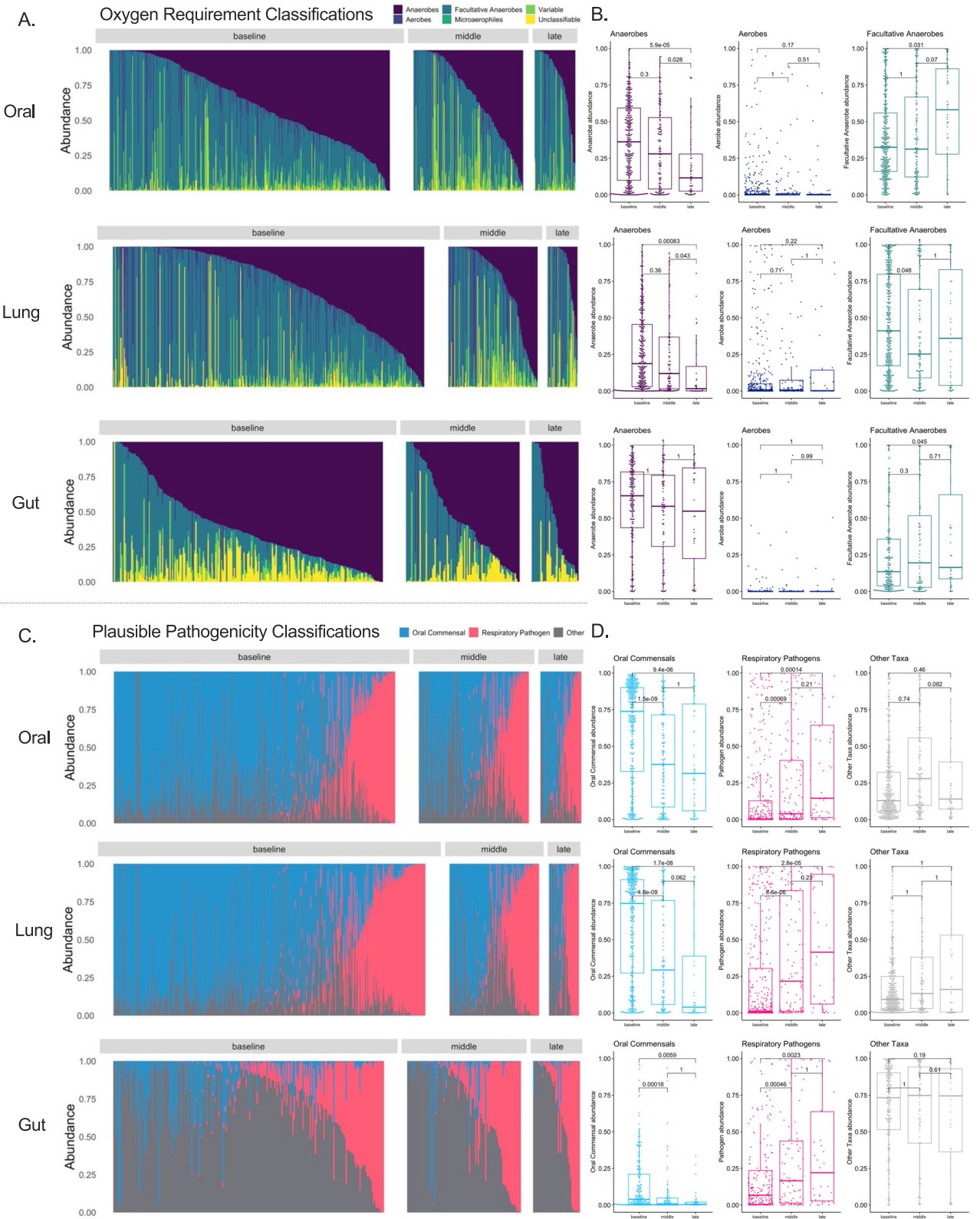

HR = 2.22 [1.07–4.63], $p$ = 0.03). Furthermore, by examining the longitudinal evolution of lung DMM clusters, patients who remained in the low diversity cluster from the baseline to the middle interval ("Low Diversity Persisters", Fig. 4) had significantly worse survival than other patients with available follow-up samples (age-adjusted HR = 2.73 [1.19–6.42], $p$ = 0.02). Thus, we found evidence that lung microbiota dysbiosis predicted survival beyond the information provided by

clinical predictors, commonly used organ dysfunction indices, and biological subphenotyping.

### Derivation of a dysbiosis index and external validation in patients with COVID-19

Motivated by the robust, independent prognostic impact of microbiota clusters on patient survival, we constructed predictive models to

**Fig. 2 | Longitudinal analysis of bacterial composition showed a progressive loss of obligate anaerobes in oral and lung communities as well as enrichment for recognized respiratory pathogens in all three compartments.** Top Panels (**A**, **B**): Relative abundance barplots for oral, lung, and gut samples with the classification of bacterial genera by oxygen requirement into obligate anaerobes (anaerobes), aerobes, facultative anaerobes, microaerophiles, genera of variable oxygen requirement and unclassifiable. Comparisons of centered-log ratio (CLR) transformed relative abundances for the three main categories of bacteria (obligate anaerobes, aerobes, and facultative anaerobes) by follow-up interval (baseline, middle and late). Data in boxplots (**B**) are represented as individual values of untransformed relative abundances, with their median as the line inside the box, interquartile range (25th–75th percentile) as the box itself, whiskers extend to 1.5 times the interquartile range, and individual dots beyond whiskers signify outlier observations. Comparisons between intervals were done by non-parametric Wilcoxon tests, with $p$-values adjusted for multiple comparisons by the Bonferroni method. Bottom Panels (**C**, **D**): Relative abundance barplots for oral, lung and gut samples with the classification of bacterial genera by plausible pathogenicity into oral commensals, recognized respiratory pathogens, and "other" category. Comparisons of CLR-transformed relative abundances for these categories of bacteria by follow-up interval (baseline, middle, and late) in boxplots (**D**), with $p$-values adjusted for multiple comparisons. Source data are provided as a Source Data file. Displayed data include 583 oral, 543 lung, and 343 gut samples from ICU patients. All statistical tests were two-sided.

classify bacterial profiles into the corresponding DMM clusters within each compartment. Such predictive models could serve as dysbiosis indices beyond the derivation cohort with our DMM analysis. We used probabilistic graphical modeling (PGM)[29] to predict the DMM clusters in each compartment based on the CLR-transformed abundance of the top 50 taxa and the corresponding Shannon index. By splitting the dataset into training and testing subsets (80% and 20% of data points, respectively), we developed separate multinomial regression models for DMM cluster predictions in each compartment (i.e. compartment-specific Dysbiosis Index), which showed accuracy of 0.80, 0.80 and 0.82 for oral, lung and gut clusters, respectively. We verified that patients classified in the low diversity clusters by the Dysbiosis Index for the oral and lung compartments had worse survival, similarly to the DMM-derived clusters.

We next applied the derived Dysbiosis Indices to two independent cohorts of hospitalized patients with COVID-19 pneumonia. In the UPMC-COVID cohort of patients with COVID-19 ARDS on IMV ($n = 49$), the Lung Dysbiosis Index classified ETA samples into three clusters with significant differences in Shannon index and bacterial load by qPCR (Fig. 5A), but no difference in ETA SARS-CoV-2 viral load by qPCR. Patients assigned to the low diversity cluster at baseline had higher ETA levels of sTNFR1, as well as higher plasma Ang-2 compared to the high diversity cluster ($p < 0.05$, Fig. 5B). By individual taxa abundance, typical pathogen abundance was correlated with intensified ETA inflammation (e.g. *Klebsiella* correlated with higher levels of ETA sTNFR1 and IL-6), several oral commensals were correlated with higher ETA levels of sRAGE (such as *Streptococcus*, *Rothia* and *Veillonella*) potentially indicating higher degree of lung epithelial injury, whereas *Prevotella* abundance was inversely correlated with plasma levels of inflammatory and tissue injury biomarkers (Figure S12). Notably, low diversity cluster patients at baseline had numerically worse outcomes of liberation from mechanical ventilation and 60-day survival, although these effects did not reach statistical significance.

In the MGH-COVID cohort ($n = 97$), we performed Illumina NovaSeq metagenomic sequencing in longitudinal lung (ETA for patients on IMV or expectorated sputum in spontaneously breathing patients) and gut (stool) samples obtained upon enrollment and then daily up to day 4. We found no significant changes over time in Shannon Index and anaerobe/pathogen abundance in either compartment on serial samples through day 4 (data not shown). We classified baseline lung and gut samples by our Dysbiosis Index models, which showed significant differences in Shannon index and anaerobe abundance in lung samples (Fig. 5D, E). Importantly, the low diversity cluster in both the lung and gut compartment was strongly associated with COVID-19 pneumonia severity classified by oxygen support requirements (odds ratios 18.07 [1.92–992.5] and 4.08 [1.56–11.2], for lung and gut clusters, respectively Fig. 5F-G). Thus, the application of the Dysbiosis Indices to lung and gut samples of patients with COVID-19 provided similar findings to the ones obtained in the UPMC-ARF derivation cohort, supporting the predictive value of microbiota profiling.

## Discussion

We conducted a longitudinal, integrative assessment of host-microbiota interactions in a large cohort of ARF patients across three body sites (the oral cavity, lungs, and gut) and up to three-time points in the ICU. These analyses offered insights into the temporal relationships between patient-level factors, therapeutic interventions, microbial communities and patient-centered outcomes, which has not been possible in previous smaller scale investigations[30]. The progressive dysbiosis of microbial communities observed in all three body compartments highlights the impact of critical illness on the global microbiota. We found reduced alpha diversity and deviation in composition compared to healthy controls at the onset of IMV, with further reduction in diversity and alterations in composition for patients supported on ventilators over time. Unsupervised analyses of microbiota composition revealed distinct communities in all three body compartments, yet the lung microbiome emerged as the strongest independent predictor of important clinical outcomes. We developed parsimonious models for dysbiosis classifications in each compartment and found that lung dysbiosis was significantly associated with host-response profiles and clinical severity in patients with COVID-19.

The large sample size and granular clinical data in our derivation cohort allowed for a detailed investigation of the relationships between patient-/treatment-related factors with the composition of microbiota. Clinical diagnoses (e.g. pneumonia) and comorbidities explained variation in diversity and composition at baseline. Despite the self-evident biological plausibility of antibiotic pressures on altering the microbiomes of critically ill patients, empirical evidence to date has been limited[21,31,32]. Here we modeled antibiotic exposure thoroughly with different methodologies from prior studies focused on cystic fibrosis or pneumonia[25,26,33], and studied antibiotic effects on longitudinal communities and features of dysbiosis. We found that the NAT score and a simple categorical classification with regard to anaerobic spectrum coverage captured important effects on longitudinal composition. Recent epidemiologic and molecular evidence supports the disruptive effects of anti-anaerobic antibiotics in gut microbial communities[21,22]. Our data are consistent with the idea that anaerobe-targeting antibiotics are associated with anaerobic bacteria depletion in the respiratory and intestinal tracts, and furthermore, our study suggests that such depletion is associated with worse clinical outcomes. Our results thus highlight the importance of rational use of anti-anaerobic antibiotics[34], as directed by proper clinical indications, because such antibiotics can have important yet under-recognized adverse clinical implications.

The biogeography of the intubated respiratory tract has been extensively investigated for prevention of secondary ventilator-associated pneumonia (VAP)[35,36]. Clinical trials have examined oropharyngeal decontamination with chlorhexidine rinses or the more aggressive selective digestive decontamination (SDD) of the gastrointestinal tract as means of reducing bacterial burden in the probable source compartments that seed the LRT microbiota. While randomized clinical trials have shown that both decontamination

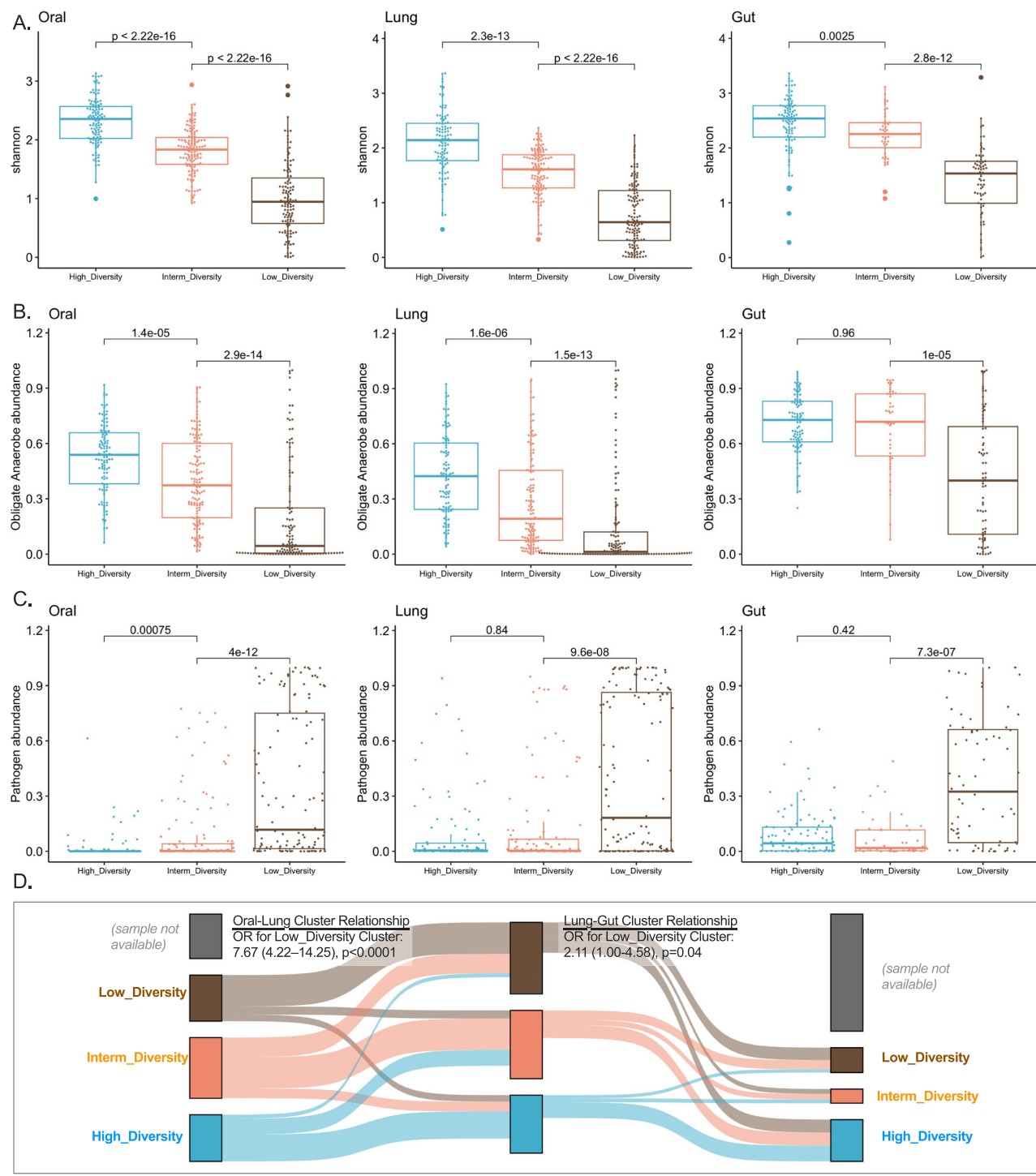

**Fig. 3 | Unsupervised clustering approaches revealed differences in bacterial alpha diversity and composition in three body compartments of critically ill patients.** Panels **A**–**D** demonstrate bacterial Dirichlet Multinomial Mixture (DMM) modeling results for each compartment separately. DMM clusters had significant differences in alpha diversity (A, Shannon index, derived from all reads in each sample) and composition (obligate anaerobe relative abundance in shown in panel **B** and pathogen relative abundance shown in panel **C**, with comparisons performed in abundances post centered-log ratio transformation), with cluster 3 in each compartment showing very low Shannon Index and enrichment for pathogens (Low-Diversity cluster). Oral and lung cluster assignments were strongly associated with each other (Odds ratio for assignment to the Low-Diversity cluster: 7.67 (422–14.25), Fisher's test $p < 0.0001$), whereas membership to lung and gut clusters was associated significantly with borderline statistical significance (Fisher's test $p = 0.04$, panel **D**). Source data are provided as a Source Data file. Displayed data include 380 oral, 393 lung, and 216 gut samples from ICU patients obtained at baseline. Data displayed as boxplots with individual dots have their median as the line inside the box, interquartile range (25th–75th percentile) as the box itself, whiskers extend to 1.5 times the interquartile range, and individual dots beyond whiskers signify outlier observations.

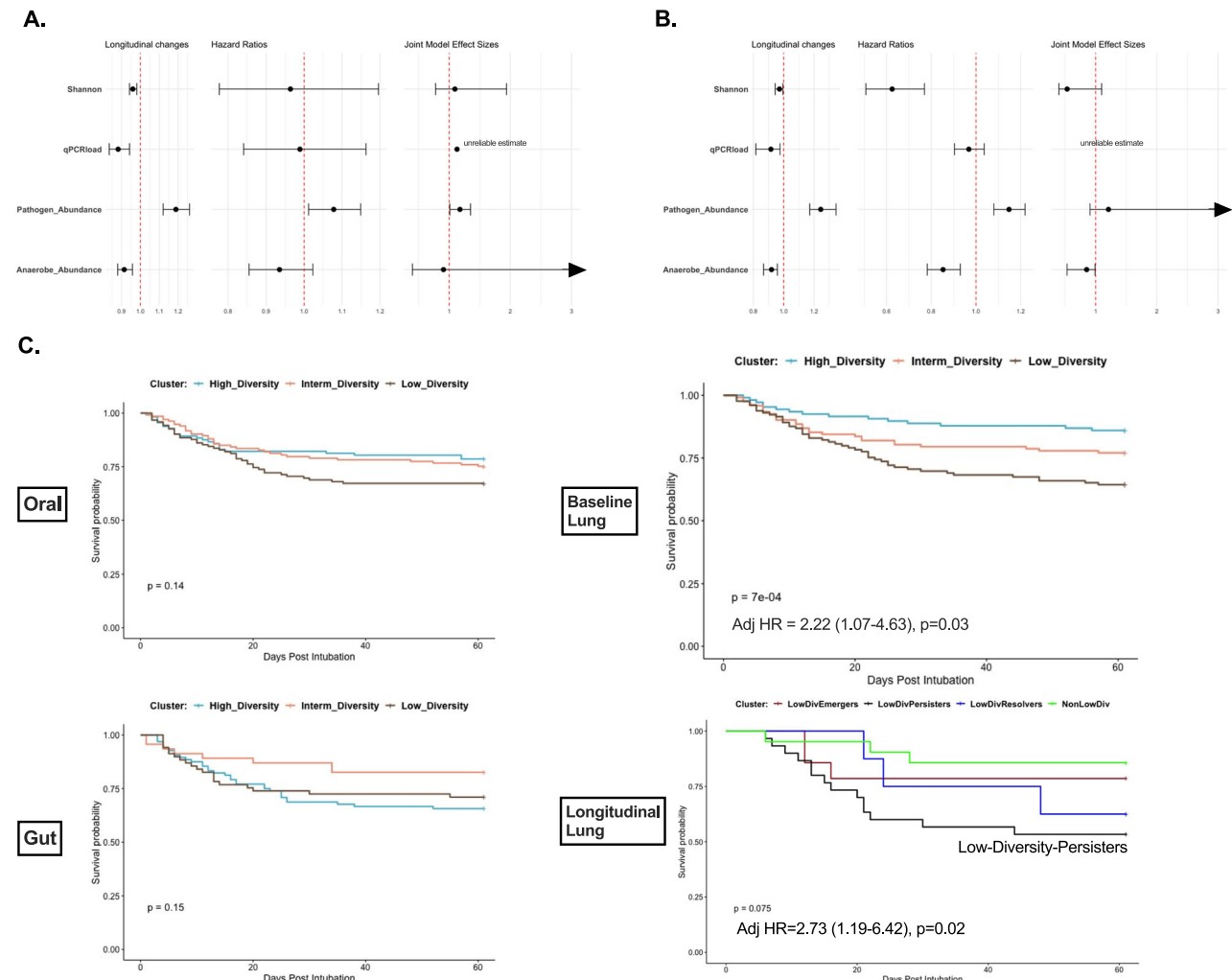

**Fig. 4 | Lung dysbiosis features and clusters predict 60-day survival. A**, **B**: Forest plots of effect sizes (point estimates and 95% confidence intervals) for dysbiosis features (Shannon index, bacterial load, anaerobe and pathogen abundance) in three different models: (i) mixed linear regression models with random patient intercepts for the longitudinal change of dysbiosis features during follow-up sampling, (ii) the age-adjusted hazards ratios from Cox-proportional hazards models for the baseline values of each feature on 60-day survival, and (iii) joint-modeling with adjusted beta-coefficient for the effect of each longitudinally-measured feature on survival. Joint modeling showed that pathogen abundance in the oral compartment and anaerobe abundance in the lung compartment had borderline statistically significant effects on 60-day survival. Joint-models for bacterial load by qPCR did not converge due to low number of longitudinal measurements. C. Kaplan-Meier curves for 60-day survival from intubation stratified by oral (**A**), lung (**B**) and gut (**C**) bacterial DMM clusters. The Low-Diversity lung DMM cluster was independently predictive of worse survival (adjusted Hazard Ratio = 2.22 (1.07–4.63), Cox regression $p$ = 0.03), following adjustment for age, sex, history of COPD, immunosuppression, severity of illness by sequential organ failure assessment (SOFA) scores and host-response subphenotypes. Longitudinal analysis of lung DMM clusters showed that patients who remained in the low diversity cluster from the baseline to the middle interval ("Low Diversity Persisters") had significantly worse survival than other patients with available follow-up samples (age-adjusted HR = 2.73 [1.19–6.42], Cox regression $p$ = 0.02). Source data are provided as a Source Data file. Displayed data include 380 oral, 393 lung and 216 gut samples from ICU patients obtained at baseline.

approaches are effective in preventing VAP[37,38], safety concerns[39,40] have led to limited uptake of SDD worldwide and de-adoption of chlorhexidine rinses. Indiscriminate application of chlorhexidine rinses in all patients on IMV may in fact deplete commensal organisms from the URT and reduce colonization resistance against pathogens. We found significant correlations between the abundance of oral-origin commensals, such as *Prevotella*, in URT and LRT samples, with lower levels of plasma inflammatory biomarkers, suggesting a potential regulation of innate immunity by such taxa[41–43]. Our comparative analyses between compartments revealed a much higher similarity between the oral and lung microbiota compared to the similarity lung and gut microbiota. This suggests that the oral cavity is the main source of microbial seeding in the lungs. However, we did observe a small subset of patients who had enrichment for gut-origin commensal or pathogenic organisms in their LRT. Such enrichment could not be

fully accounted for by URT colonization with similar taxa. These patients with gut-origin bacteria enrichment in their lungs (8.3%) had much worse survival than the rest of the cohort. This subset of patients may have experienced gut-to-lung bacterial translocation[19,20]. To further investigate the possibility of gut-to-lung translocation, it would be beneficial to have a wider availability of BAL samples to investigate the alveolar spaces more closely. Our non-invasive ETA samples showed that such translocation, if present, affects a small subset of patients at least within the first week of IMV. Therefore, efforts focused on preventing lung dysbiosis and pathogen colonization will need to consider primarily the URT-to-LRT ecosystem and secondarily the possibility of gut-to-lung translocation.

Unsupervised clustering revealed distinct microbial communities within and across body compartments. Low-diversity bacterial clusters were enriched with pathogens and depleted in anaerobes in all three

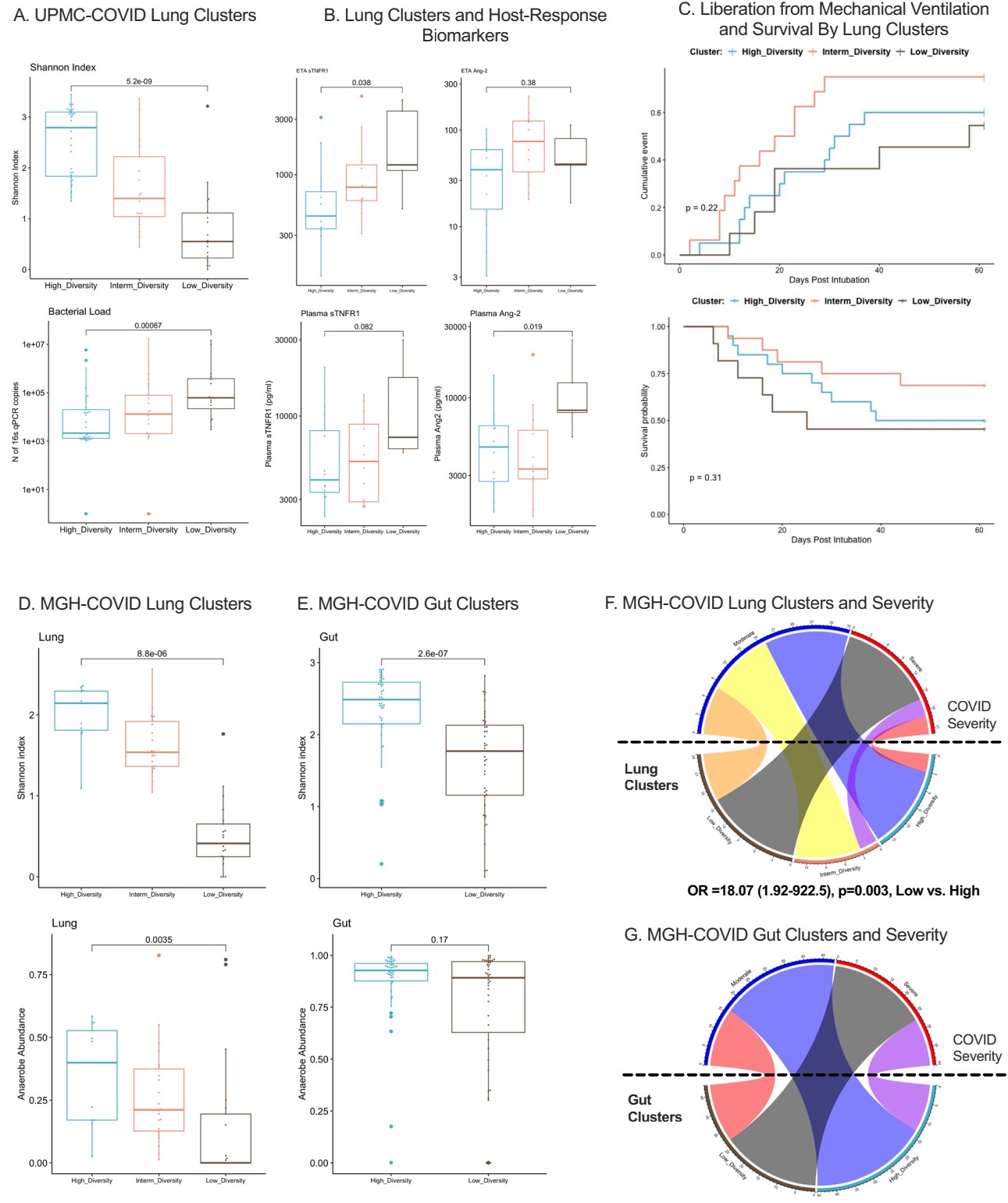

compartments. Membership in the low-diversity cluster was strongly associated with the oral and lung compartments, suggesting shared patterns of dysbiosis. The overall stability of longitudinal cluster membership indicated that specific microbial profiles may persist throughout critical illness, influencing the disease trajectory. Integration of fungal sequencing data further enhanced our view of the microbial communities, revealing patients who had enrichment for *C.albicans* and experienced worse outcome. The presence of

*C.albicans* in the LRT may not signify clinical pneumonia by conventional criteria[44], yet may represent a state of dysbiosis with potential adverse effects from *C.albicans* on host epithelial integrity and immune response. Taxonomic concordance between amplicon sequencing approaches (Illumina 16S-Seq and ITS-Seq) with rapid metagenomic sequencing with the MiNION device (Oxford Nanopore Technologies, UK) suggests that such microbial profiles could be generated in clinically relevant turnaround times[45].

**Fig. 5 | Lung and Gut Microbiota Associations with COVID-19 Severity in Two Independent Cohorts. A** Application of the dysbiosis index in lung (ETA) microbiota profiles in the UPMC-COVID cohort classified subjects in three clusters, with significant differences in Shannon index and bacterial load by 16S qPCR. **B** The low diversity cluster in lung samples from UPMC-COVID subjects was significantly associated with higher ETA levels of sTNFR1 and plasma levels of Ang-2. **C** COVID-19 patients classified to the low diversity cluster had numerically worse time-to-liberation from invasive mechanical ventilation and survival. **D, E** Application of the dysbiosis index models in lung (sputum or ETA) and gut (stool) samples in the MGH-COVID cohort classified subjects in three clusters, with significant differences in Shannon index and anaerobe abundance between clusters. **F, G** Cluster assignments in the MGH cohort were strongly associated with clinical severity. Membership in the Low-Diversity cluster in the lungs was associated with an odds ratio of 18.07 (1.92-922.5) for severe disease (black belt connecting the Low-Diversity cluster and Severe Disease perimetric zones in the chord diagram). Membership in the low diversity gut cluster was also significantly associated with clinical severity of COVID-19 pneumonia (odds ratio of 4.08 [1.56–11.2]). Source data are provided as a Source Data file. Displayed data include 47 baseline lung samples from UPMC-COVID, and 75 baseline lung and 88 stool samples from MGH-COVID cohort. Data displayed as boxplots with individual dots have their median as the line inside the box, interquartile range (25th–75th percentile) as the box itself, whiskers extend to 1.5 times the interquartile range, and individual dots beyond whiskers signify outlier observations.

Survival analyses based on microbiota clusters yielded significant findings. First, when comparing microbiota from three distinct body compartments to predict survival, the lung microbiome emerged as the strongest predictor compared to oral or gut microbiota. This finding is not surprising given that previous research had shown that baseline lung microbiota profiles were predictive of survival[3]. Expanding on these prior observations, we analyzed microbiota in three compartments at various time points during IMV and found that the lung microbiota continued to be the most predictive, both at baseline and also in follow-up samples. Notably, we modeled the longitudinal change in dysbiosis features and its impact on survival with joint modeling. Joint modeling is a flexible approach that can mitigate some of the effects of informative censoring[28]. The latter is particularly relevant for translational studies in the ICU because patients who experienced early mortality or those who improved quickly and were discharged from the ICU could not contribute later follow-up samples[46]. Our joint models revealed that baseline features in the lung compartment (Shannon index and anaerobe or pathogen abundance) were predictive of survival, whereas their longitudinal changes were not, except for marginal effects of anaerobe abundance. Such results may indicate that the communities formed by host-microbiota interactions early post-intubation are already representative of an LRT infection or dysbiosis state. Therefore, subsequent changes among those who remain intubated in the ICU may be less consequential for the overall outcome compared to their starting state. Nonetheless, longitudinal observations were limited by a lower number of observations, which has likely limited the statistical power of joint models, as indicated by the wide CI in effect estimates. Overall, our inter-compartment comparisons highlight the clinical relevance of lung microbiota analysis in critical illness, and underscore the need for dedicated sampling of the LRT[47].

The prognostic value of lung microbiota clusters was independent not only from clinical predictors and validated organ dysfunction metrics, such as the SOFA score but also from the systemic host-response subphenotypes. Extensive evidence has established the prognostic value and generalizability of plasma biomarker-based subphenotyping of patients with ARF[8,48]. Hyperinflammatory patients had higher pathogen abundance in lung samples compared to hypoinflammatory patients. Notably, the hyperinflammatory phenotype has been associated with a higher burden of bacteremia and circulating microbial cell-free DNA[49,50]. Our adjusted Cox proportional hazards models revealed significant hazard ratios for the Low-Diversity lung cluster. Beyond the significant taxa-biomarker associations we observed, these survival analyses demonstrate that lung microbiota may influence patient outcomes in ways that are not captured by current host-response subphenotyping approaches. An integrative, host- and lung microbiome-aware subphenotyping framework may thus augment our ability to better prognosticate and target therapeutic interventions in ARF[51].

Our study has several limitations. First, we mainly focused on bacterial and fungal components of the microbiome, and thus could not assess the role of the virome, especially with regards to respiratory RNA viruses. The consistent pattern of results relating elements of the bacterial microbiome to host response and illness severity in the COVID-19 cohorts supports the generalizability of our findings, although we could not investigate contributions from individual viruses. The observational nature of our study prevents us from establishing causality between the microbiome and clinical outcomes, which could be addressed by future interventional studies or animal modeling with microbiome manipulation. Longitudinal sample availability was limited by informative censoring, which we attempted to mitigate with mixed linear regression and joint modeling approaches. Our longitudinal analysis findings should be interpreted with caution and considered as applicable to patients who remain on IMV for the first 1–2 weeks of critical illness.

For patient safety and practical purposes of subject participation in our observational research study, we relied on non-invasive biospecimens (ETA) for LRT microbiota profiling, as opposed to reference standard BAL[47]. Our non-invasive approach allowed us to enroll a large cohort of LRT specimens, follow serial samples over time and is congruent with clinical practice guidelines for VAP diagnosis[52]. Recent research has shown the ability to derive robust microbiota signatures from ETA samples in patients on IMV[53]. However, we may have missed important microbiota variability closer to the alveolar space, including potentially stronger deviation from URT microbiota, a higher signal of gut-to-lung microbiota translocation, as well as better delineation of longitudinal host-response biomarkers in BAL fluid[20]. In a limited comparison of two subjects with synchronous ETA-BAL sampling, we found that in a case of *Achromobacter xylosidans* pneumonia, both ETA and BAL samples showed community dominance (>90% relative abundance) by *Achromobacter* genera, whereas in a case of culture-negative pneumonia diagnosis, taxonomic concordance between ETA and BAL sample was more limited. These results are consistent with a previous comparison of ETA vs. mini-BAL metagenomics, in which case higher taxonomic concordance was seen for cases with culture-positive pneumonia[54]. Thus, the reliability of ETA biospecimens for profiling airspace microbiota may be context dependent, and further research is needed with BAL biospecimens when available[55]. Additionally, we had a smaller effective sample size for gut microbiota analysis, which may have limited our ability to identify prognostic variation within the gut compartment. Last, our derivation cohort had limited racial/ethnic diversity consistent with the demographics of the catchment populations for our ICUs; therefore, our results require independent validation in more diverse patient populations.

In conclusion, our study provides important insights into the predictive value of microbiota clusters derived from different body compartments in critically ill patients. The lung microbiome emerged as the most powerful predictor of survival, surpassing the oral and gut microbiota. These findings emphasize the clinical relevance of investigating the lung microbiota and highlight its potential as a prognostic marker in critical illness. Moreover, our study underscores the importance of considering organ-specific microbial communities in critical care settings and expands our understanding of the microbiome's role in determining patient outcomes. Further research in this area has the potential to shape clinical decision-making and facilitate

the development of personalized medicine strategies for critically ill patients.

## Methods

Our research complies with all ethical regulations in accordance with the Declaration of Helsinki, and as directed by the University of Pittsburgh Institutional Review Board (IRB) (protocols STUDY19050099, STUDY19060243, and STUDY20060312) and the Mass General Brigham IRB (protocol #2020P000804).

### UPMC-ARF cohort

Following admission to the ICU at UPMC (Pittsburgh, PA, USA) and obtaining informed consent from patients or their legally authorized representatives (University of Pittsburgh IRB protocol STUDY19050099), we collected baseline research biospecimens within 72 hours from intubation. We collected blood for separation of plasma, oropharyngeal (oral) swabs to profile upper respiratory tract (URT) microbiota, endotracheal aspirates (ETA) for LRT (lung) microbiota and host biomarker measurements, and rectal swabs or stool samples for gut microbiota analyses. We also captured leftover bronchoalveolar lavage fluid (BALF) from clinically indicated bronchoscopies, when available. We repeated research biospecimen sampling between days 3–6 (middle interval) and days 7–12 (late interval) post-enrollment for subjects who remained in the ICU. No patients in the UPMC-ARF cohort were known to be infected by SARS-CoV-2 at the time of enrollment.

### UPMC-COVID cohort

Following admission to the ICU and obtaining informed consent from patients or their legally authorized representatives (University of Pittsburgh IRB protocol STUDY19050099), we collected baseline research biospecimens (ETA and blood) within 72 hrs from intubation. We repeated research biospecimen sampling between days 3-6 (middle interval) and days 7–12 (late interval) post-enrollment for subjects who remained in the ICU, as per the UPMC-ARF protocol. All patients were known to have positive SARS-CoV-2 qPCR prior to enrollment.

### MGH-COVID cohort

From April 2020 to May 2021, we prospectively enrolled 97 hospitalized patients aged ≥18 years with confirmed COVID-19 at the Massachusetts General Hospital (Boston, MA, USA) in a longitudinal COVID-19 disease surveillance study[15]. The Study protocol #2020P000804 was approved by the Mass General Brigham IRB. All participants or their healthcare proxy provided written informed consent to participate. Patients were categorized as having severe COVID-19 if they required admission to the intensive care unit with acute respiratory failure (the need for oxygen supplementation ≥15 liters per minute (LPM), non-invasive positive pressure ventilation, or mechanical ventilation) or other organ failure (such as shock requiring vasopressors). Otherwise, they were categorized as having moderate COVID-19. Expectorated sputum, ETA, or fresh stool was collected and refrigerated at 4 °C until aliquoting/freezing at −80 °C (typically within 4 hours of collection) from adult patients enrolled in the prospective biospecimen collection study. Participants were able to provide samples as frequently as once daily for up to four days, as well as declining donations on any given day (while remaining in the study).

### Healthy controls

To contextualize the findings on microbiota from critically ill patients with what is expected for the healthy respiratory and gastrointestinal tract, we also included data from 24 healthy volunteers who had contributed URT and LRT microbiome data in a previously published cohort (Lung HIV Microbiome Project – University of Pittsburgh IRB STUDY19060243)[56], as well as stool from 15 healthy donors for fecal microbiota transplantation (University of Pittsburgh IRB -

STUDY20060312)[11]. We designated these healthy volunteers as Healthy Controls.

### Clinical data recording

We obtained clinical data directly from the electronic medical record. We captured biological sex and race as recorded in the medical record. A consensus committee reviewed clinical and radiographic data and performed retrospective classifications of the etiology and severity of acute respiratory failure without knowledge of microbiome sequencing or biomarker data. We retrospectively classified subjects as having ARDS per established criteria (Berlin definition), being at risk for ARDS because of the presence of direct (pneumonia or aspiration) or indirect (e.g., extrapulmonary sepsis or acute pancreatitis) lung-injury risk factors although lacking ARDS diagnostic criteria, having acute respiratory failure without risk factors for ARDS, or having acute-on-chronic respiratory failure. We followed patients prospectively for cumulative mortality and ventilator-free days (VFDs) at 30 days, as well as survival up to 60 days from intubation.

We systematically reviewed administered antibiotic therapies since hospital admission and recorded the antibiotic exposure for each subject according to the following three metrics:

1. Anaerobic coverage (yes/no): whether antibiotics with anaerobic coverage were given on the day of sampling.
2. The Antibiotic Exposure score by Zhao et al.[25]: a numerical scale with antibiotic weighting based on dosing duration, the timing of administration relative to sample collection, and the antibiotic type and route of administration. We utilized the convex increasing weighting scheme and modeled the antibiotic exposure from hospital admission until the day of sampling.
3. The Narrow Antibiotic Treatment (NAT) score was developed for community-acquired pneumonia treatment studies[26,33]. We calculated the daily NAT score from -5 days from sampling to post-10 days after sampling on day 1.

### Research sample collection

Within the first 48 hours of intubation (baseline time-point), we collected a posterior oropharyngeal (oral) swab via gentle swabbing of the posterior oropharynx next to the endotracheal tube with a cotton tip swab for 5 secs, and an endotracheal aspirate (ETA) via suctioning secretions from the endotracheal tube with the in-line suction catheter and without breaking seal in the ventilatory circuit[1,4]. Rectal swabs were collected according to a standard operating procedure (i.e., placing the patient in a lateral position, inserting the cotton tip of the swab into the rectal canal, and rotating the swab gently for 5 secs), unless clinical reasons precluded movement of the patient (e.g., severe hemodynamic or respiratory instability). Stool samples were collected when available, either by taking a small sample from an expelled bowel movement (before cleaning of the patient and disposal of the stool) or from a fecal management system (rectal tube) placed for management of diarrhea and liquid stool collection. We also collected simultaneous blood samples for centrifugation and separation of plasma. Samples were delivered to the processing laboratory within minutes from acquisition, and then aliquoted and stored in −80 °C until conduct of experiments. For samples that underwent host DNA depletion for Nanopore sequencing, an aliquot remained in 4 C for processing before freezing for up to 72 hrs from acquisition. ETA Samples obtained from COVID-19 subjects inactivated by 4-fold dilution in DNA/RNA Shield (Zymo Research) under biosafety level 2+ conditions and then stored at −80 °C. For patients who remained intubated in the ICU, we collected follow-up samples at a middle time-point (days 3–6) and a late follow-up interval (days 7–11 post-intubation).

For Healthy Controls, an oral wash and BAL sample were collected with a standardized protocol[56]. Subjects were asked to fast and refrain from smoking for at least 12 hrs before sample collection. Oral washes were performed by having participants gargle with 10 ml sterile 0.9%

saline immediately before bronchoscopy. BAL was performed according to standardized procedures developed to minimize oral contamination. Participants were gargled with an antiseptic mouthwash (Listerine) immediately before topical anesthesia. The bronchoscope was then inserted through the mouth and advanced to a wedge position quickly and without the use of suction. BAL was performed in the right middle lobe or lingula up to a maximum of 300 ml 0.9% saline. Healthy donors of stool for fecal microbiota transplant collected a stool sample in a specialized container and brought the stool sample on the day of collection to the processing lab.

## Laboratory and bioinformatics analyses

**Microbiome assays in UPMC cohorts.** From all available oral swabs, ETAs, left-over BALF, rectal swabs and stool samples, we extracted genomic DNA and performed quantitative PCR (qPCR) of the V3-V4 region of the 16 S rRNA gene to obtain the number of gene copies per sample, as a surrogate for bacterial load. From a separate aliquot of extracted DNA from oral swabs, ETA, rectal swabs, and stool samples, we performed amplicon sequencing for bacterial DNA (16S-Seq of the V4 hypervariable region) and fungal DNA (ITS) on the Illumina MiSeq platform at University of Pittsburgh Center for Medicine and the Microbiome laboratories[3,23] We used extensive experimental negative controls in all processing steps to rule out contamination, as well as mock microbial community positive controls (Zymo) to ensure target amplification success. We processed derived 16S sequences with a custom Mothur-based pipeline[57] (v1.44.1) (available at https://github.com/MedicineAndTheMicrobiome/AnalysisTools/tree/master/16S_Clust_Gen_Pipeline). In brief, we deconvoluted sequences from the Illumina MiSeq and processed them through an in-house sequence quality control pipeline, which includes dust low complexity filtering, quality value (QV < 25) trimming, and trimming of primers used for 16S rRNA gene amplification, and minimum read length filtering. Trimmed reads shorter than 75 bp or those with less than 95% of the bases above a QV of 25 were discarded. Forward and reversed paired reads were merged into contigs and processed for 16S rRNA gene sequence clustering and annotation pipeline. Sequence taxonomic classifications were performed with the Ribosomal Database Project's (RDP) naïve Bayesian classifier with the SILVA 16S rRNA database[58] (v138). ITS rRNA gene sequences from the pooled sequencing run were demultiplexed into individual sample/replicate fastq files. The variable-length reads were processed, trimmed, and quality filtered with a quality control pipeline utilizing the *DADA2* package in R, v4.2.0. Paired sequences with forward and reverse reads passing the quality filtering and trimming steps were merged. Chimeras were removed and the Unite database was utilized to classify reads into amplicon sequence variants (ASVs) using the naïve Bayesian classifier method, defined at the species level.

From a random subset of 130 available ETA samples, we performed metagenomic Nanopore sequencing (following human DNA depletion) with a rapid PCR barcoding kit (SQK-RPB004) on the MinION device (Oxford Nanopore Technologies-ONT, Oxford, UK) for six hours[59,60]. We analyzed microbial metagenomic sequences with the EPI2ME platform (ONT) and the "What's In My Pot" [WIMP] workflow to quantify the abundance of microbial species[61]. We filtered FASTQ files with a mean quality (q-score) below a minimum threshold of 7.

**Host-response assays.** We measured 10 plasma biomarkers of tissue injury and inflammation with custom Luminex multi-analyte panels from plasma samples and ETA supernatants, when available. Specifically, we used a 10-plex Luminex panel (R&D Systems, Minneapolis, MI, United States) to measure interleukin(IL)-6, IL-8, IL-10, soluble tumor necrosis factor receptor 1 (sTNFR1), suppressor of tumorigenicity-2 (ST2), fractalkine, soluble receptor of advanced glycation end-products (sRAGE), angiopoietin-2, procalcitonin and pentraxin-3[7].

## Microbiome assays in MGH-COVID cohort

Samples were extracted and sequenced at Baylor College of Medicine according to their standard established platforms. DNA was prepared for sequencing using the Illumina Nextera XT DNA library preparation kit. All libraries were sequenced with a target of 3 GB output at 2 x 150 bp read length using the Illumina NovaSeq platform[15]. Taxonomic profiles were generated using the bioBakery 3 shotgun metagenome workflow 3.0.0, the details of which have previously been described[62]. Briefly, human reads were filtered using KneadData 0.10.0, and species-level taxonomic profiles were generated using MetaPhlAn 3.0.0[63].

## Quantification and statistical analysis

We performed non-parametric comparisons for continuous (described as median and interquartile range – IQR) and categorical variables between clinical groups (Wilcoxon and Fisher's exact tests, respectively). For microbial community profiling, we included samples that produced ≥1000 high-quality microbial reads for 16S-Seq. In the UPMC-ARF cohort, we considered 1520 unique clinical samples (593 oral swabs, 578 ETA/BALF [lung], and 349 stool or soiled rectal swabs [gut]). Filtering at 1000 reads resulted in elimination of 112 clinical samples (24 oral swabs, 77 lung samples, and 11 gut samples). We performed 16S-Seq analyses at the genus level, which were filtered for singletons and low abundance taxa (i.e. those with relative abundance <0.0001 in <5% of samples), resulting in a final set of 214 unique genera for analyses. We analyzed ITS-Seq at species level for fungi, and Nanopore metagenomics at species level for DNA reads of bacteria, fungi, and viruses. We calculated alpha diversity with the Shannon index following rarefaction at 1000 reads, with 100 random subsamplings to obtain the average Shannon index for each available sample. Rarefaction curves showed that all clinical samples had reached a plateau for calculation of the Shannon index by 1000 reads (Figure S1). We applied rarefaction to allow for between-sample type comparisons because sequencing depth and yield varied between sample types. We conducted between-group comparisons of alpha diversity with two-sided non-parametric tests at a significance level of 0.05 to draw inferences on systematic differences of alpha diversity between groups as a measure of relative community fitness[1]. We conducted beta diversity analyses (Bray-Curtis indices) on centered log-ratio (CLR) transformed abundances, analyzed via permutation analysis of variance (Permanova), and visualized via principal coordinates analyses with the R *vegan* and *mia* packages[64]. We adjusted all reported *p*-values from Permanova tests for multiple testing with a conservative Bonferroni correction. We examined for differentially abundant taxa between groups following CLR transformations with the *limma* package to fit weighted linear regression models, performed tests based on an empirical Bayes moderated *t*-statistic, and obtained False Discovery Ratio corrected *p*-values. We examined for correlations between CLR-transformed abundances and host response biomarkers with the Pearson's correlation co-efficient. Examination of the taxonomic composition of negative control samples revealed very low numbers of reads for commonly detected taxa (Figure S1). Examination of the taxonomic composition of negative control samples revealed very low numbers of reads for commonly detected taxa (mean <100 reads, Figure S1) and we did not filter any taxa from clinical samples.

We then examined the discovered bacterial taxa at the genus level and classified them by two different classification schemes with clinical relevance. First, we considered the oxygen requirements of each bacterial taxon, given the relevance of oxygen metabolism in critically ill patients on invasive mechanical ventilation who receive variable amounts of inspired oxygen. Recent research has investigated the impact of hyperoxia in LRT microbiota[65], as well as the association between anaerobic spectrum antibiotics with anaerobe bacteria abundance in the respiratory and gastrointestinal tract[12]. We thus

classified the analyzed taxa by oxygen requirements for bacterial metabolism (details in Table S3):

1. Obligate aerobes (referred to throughout as aerobes): bacteria that require oxygen to grow and survive, as they use oxygen as a final electron acceptor in their respiratory chain.
2. Obligative anaerobes (referred to throughout as anaerobes): bacteria that are unable to grow in the presence of oxygen, as they are unable to use oxygen as a final electron acceptor and are killed in the presence of oxygen.
3. Facultative anaerobes: bacteria that can grow in the presence or absence of oxygen. They can use both aerobic and anaerobic respiration, depending on the availability of oxygen in their environment, switching from aerobic to anaerobic metabolism.
4. Microaerophiles: bacteria that require a low level of oxygen to grow and survive, as they can grow at oxygen concentrations lower than those required by obligate aerobes but higher than those tolerated by obligate anaerobes.
5. Variable: genera that included both aerobes and anaerobes and could not be classified further with confidence.
6. Unclassifiable: taxa that were not classified at the genus or family level with confidence to allow assessment of their metabolic needs.

Next, we classified organisms based on their plausible pathogenicity. Prior research has examined the associations between oral-origin bacteria in the LRT (i.e. lung commensals) with innate immunity and clinical outcome[42], as well as the importance of detecting highly abundant typical respiratory pathogens as causal factors of LRT infection in critically ill patients[3,5]. We therefore utilized operational definitions of plausible pathogenicity of detected taxa as follows (details in Table S4):

1. Common respiratory pathogens: bacteria considered to be typical pathogens when isolated in LRT microbiologic cultures.
2. Oral-origin commensal bacteria: bacterial taxa that have been characterized as typical members of the lung microbiome in health and originate from the oral cavity.
3. Other: taxa with unclear clinical significance that do not fall into categories 1 or 2 above.

To agnostically examine our samples for distinct clusters of microbial composition ("metacommunities"), we applied unsupervised Dirichlet multinomial models (DMMs). We used Laplace approximations[66] to define the optimal number of clusters in our dataset, and a prevalence criterion, requiring that each additional cluster would contain ≥10% of observations to merit clinical relevance for inclusion. We then examined for associations between DMM clusters with clinical parameters and outcomes. We classified subjects into a hyper- vs. hypo-inflammatory subphenotype based on predictions from a parsimonious logistic regression model utilizing plasma levels of sTNFR1, Ang-2, and procalcitonin (research biomarkers measured with Luminex panel), as well as serum bicarbonate levels measured during clinical care.

We followed patients prospectively and constructed Kaplan-Meier curves and Cox-proportional hazard models for 60-day survival, adjusted for the predictors of age and sex, as well as plausible confounders of microbiome associations diagnosis based on our findings (history of COPD, history of Immunosuppression), severity of illness as per the SOFA score, and host-response subphenotypes. To examine the impact of mechanical ventilation, steroids, and antibiotics pressure on longitudinal microbiota profiles, we constructed mixed regression models with random patient intercepts and adjusted for the number of days post-intubation that each sample was taken (as a proxy for the exposure to the hyperoxic environment of the ventilator) and the antibiotic exposure and steroids metrics by the day of sampling.

In each body compartment (oral, lung, and gut), we examined for the impact of dynamic changes in microbiota features (rarefied Shannon Index, bacterial load by 16S qPCR, CLR-transformed Anaerobe abundance, and CLR-transformed Pathogen abundance) on 60-day survival using joint modeling. The joint models combined a mixed linear regression model with random patient intercepts for measuring the longitudinal changes of each feature during sampling follow-up, and a Cox-proportional hazards model adjusted for age. Joint modeling offers the advantage of providing estimates of time-related associations with outcome and can handle informative censoring, which may have impacted our follow-up sample availability (e.g. in the case of patients with early mortality or patients with rapid clinical improvement and discharge from the ICU). We built joint models with the *joineR* package, and reported in graphical format: i) the beta-coefficients with 95% confidence intervals (CI, estimated via bootstrapping at 100 iterations) for the longitudinal change of the microbiota variables during follow-up from the mixed linear regression models, ii) the age-adjusted hazards ratios with 95% confidence intervals from Cox-proportional hazards models for the baseline values of each microbiota variable on 60-day survival, and iii) the joint-modeling adjusted beta-coefficient for the effect of each variable on survival. We performed all statistical analyses in R v.4.2.0[67].

Following the derivation of the DMM clusters in each compartment of the UPMC-ARF cohort and the demonstration of significant associations with patient outcomes, we proceeded to develop multinominal logistic regression models for the prediction of classification of bacterial 16S profiles from new samples into predicted cluster assignments. We considered these new classification models as a Dysbiosis Index for each compartment. To develop these models in each compartment (oral, lung, and gut), we used probabilistic graphical modeling (PGM)[68] by considering the 50 most abundant taxa (expressed by CLR-transformed relative abundance) in each compartment along with the Shannon Index. We divided the samples of each compartment into two random subsets: 80% of data points for training and 20% for testing. The training set was used to generate a PGM using the FCI-MAX algorithm with an Alpha of 0.1 to examine which variables (50 taxa abundance and Shannon Index) were associated with the cluster assignments in each compartment. The variables that appeared in the Markov blanket of the DMM cluster assignment variable were used to create a multinomial logistic regression (MLR) model to predict the cluster assignment of future samples. The MLR model equations were written as follows for the different cluster assignments (Low, Intermediate, and High Diversity):

Model equations

$$\ln\left(\frac{P(Intermediate)}{P(HighDiversity)}\right) = b_{10} + b_{11}f_1 + \ldots + b_{1n}f_n \tag{1}$$

$$\ln\left(\frac{P(LowDiversity)}{P(HighDiversity)}\right) = b_{20} + b_{21}f_1 + \ldots + b_{2n}f_n \tag{2}$$

$$f : feature$$

$$b1 \, \& \, b2 \, are \, model \, coefficients$$

By rewriting the equations, we get the following:

$$\frac{P(Intermediate)}{P(HighDiversity)} = e^{(b_{10} + b_{11}f_1 + \ldots + b_{1n}f_n)} \tag{3}$$

$$\frac{P(LowDiverity)}{P(HighDiversity)} = e^{(b_{20} + b_{21}f_1 + \ldots + b_{2n}f_n)} \qquad (4)$$

We rewrote the names of the model parameters as:

$$P(HighDiversity) = P(H)$$

$$P(Intermediate) = P(I)$$

$$P(LowDiverity) = P(L)$$

$$e^{(b_{10} + b_{11}f_1 + \ldots + b_{1n}f_n)} = X$$

$$e^{(b_{20} + b_{21}f_1 + \ldots + b_{2n}f_n)} = Y$$

We know that

$$P(H) + P(I) + P(L) = 1 \qquad (5)$$

Then

$$P(H) = 1 - P(I) - P(L) \qquad (6)$$

From Eqs. 3 and 4

$$P(I) = X\,P(H)$$

$$P(H) = \frac{P(L)}{Y}$$

Substituting in Eq. 6

$$\frac{P(L)}{Y} = 1 - X\frac{P(L)}{Y} - P(L)$$

$$\frac{P(L)}{Y} + X\frac{P(L)}{Y} + P(L) = 1$$

$$P(L)\left[\frac{1}{Y} + \frac{X}{Y} + 1\right] = 1$$

$$P(L)\left[\frac{1 + X + Y}{Y}\right] = 1$$

$$P(L) = \left[\frac{Y}{1 + X + Y}\right]$$

$$P(H) = \left[\frac{1}{1 + X + Y}\right]$$

$$P(I) = \left[\frac{X}{1 + X + Y}\right]$$

The predicted cluster is the one with the highest probability. For example, if $\max(P(H), P(I), P(L)) = P(I)$, then the predicted cluster is $P(I)$.

The Intercepts and coefficients for the MLR models for each compartment are provided below.

**Oral**

| f | $b_1$ | $b_2$ |
|---|---|---|
| Intercept | 3.639385 | 15.535322 |
| Pasteurellaceae_uncl | −0.4745086 | −2.3936493 |
| Alloprevotella | −0.384223 | −1.086375 |
| Anaerovoracaceae_ge | −0.5694409 | −1.0899032 |
| Centipeda | −0.5507802 | −1.7643090 |
| Fusobacterium | −0.2282283 | −1.2268803 |
| Streptococcus | 0.3062403 | −0.1848412 |
| Veillonella | −0.1108369 | −0.9535763 |

**Lung**

| f | $b_1$ | $b_2$ |
|---|---|---|
| Intercept | 6.314866 | 11.631919 |
| ShannonIndex | −2.064372 | −3.663621 |
| Porphyromonas | −0.2249019 | −1.0790556 |
| Peptostreptococcus | 0.0226691 | 0.6814249 |
| Gemella | 0.03023337 | −0.23503918 |
| Actinomyces | 0.1083389 | −0.6340821 |
| Rothia | 0.3937065 | 0.5838788 |
| Treponema | −1.120237 | −1.605860 |
| Bergeyella | −0.1974596 | −0.6481431 |
| Alloprevotella | −0.3517737 | −0.7556098 |
| Solobacterium | −0.2284961 | −0.8208410 |
| Atopobium | −0.2868991 | −0.3039660 |
| Neisseriaceae_uncl | −0.3906575 | −2.6100531 |

**Gut**

| f | $b_1$ | $b_2$ |
|---|---|---|
| Intercept | −0.469294 | 5.715458 |
| ShannonIndex | 0.05286898 | −2.57213563 |
| Campylobacter | 0.183625542 | 0.006264282 |
| Fenollaria | 0.1631794 | −0.4952786 |
| Ezakiella | 0.19868003 | −0.08232344 |
| Enterococcus | 0.5214449 | 0.6635925 |
| Blautia | −1.217537 | −0.795908 |

We tested the MLR model using three datasets: The 20% testing set for estimating model accuracy, and the ALIR-COVID samples and the MGH-COVID samples for examining associations between the Dysbiosis Index with clinical variables and endpoints.

Applications of the MLR models (Dysbiosis Index) in the three compartments showed the following accuracy statistics (95% confidence intervals) for prediction of the DMM clusters:

Oral Dysbiosis Index: 0.8026 (0.6954, 0.8851)
Lung Dysbiosis Index: 0.8 (0.6917, 0.8835)
Gut Dysbiosis Index: 0.8222 (0.6795, 0.92)

### Reporting summary

Further information on research design is available in the Nature Portfolio Reporting Summary linked to this article.

### Data availability

The Sequencing data collected for the study have been deposited to the Sequencing Resource Archive, through the following Accession numbers:-PRJNA595346 for 16S data of UPMC-ARF and UPMC-COVID cohorts, available at https://www.ncbi.nlm.nih.gov/

bioproject/PRJNA595346. -PRJNA726955 for ITS data of UPMC-ARF cohort, available at: https://www.ncbi.nlm.nih.gov/bioproject/726955- PRJNA554461 for Nanopore data of UPMC-ARF cohort, available at: https://www.ncbi.nlm.nih.gov/bioproject/PRJNA554461-PRJNA940725 for 16S data of the Healthy Controls, available at https://www.ncbi.nlm.nih.gov/bioproject/PRJNA940725-PRJNA976404 for Metagenomic data of the MGH-COVID cohort, available at https://www.ncbi.nlm.nih.gov/bioproject/PRJNA976404 De-identified clinical and processed microbiome data for the replication of analyses are available on the GitHub repository (https://github.com/MicrobiomeALIR/MultiCompartmentMicrobiome). Source data are provided in this paper for all figures and Tables. Source data are provided in this paper.

## Code availability
Primary code is available on the GitHub repository (https://github.com/MicrobiomeALIR/MultiCompartmentMicrobiome), with an archive of the code including a Digital Object Identifier available through Zenodo at https://doi.org/10.5281/zenodo.11109543.

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

## Acknowledgements

The authors wish to thank the patients and patient families who have enrolled in the University of Pittsburgh Acute Lung Injury Registry. We also thank the physicians, nurses, respiratory therapists, and other staff

at the University of Pittsburgh Medical Center Presbyterian, Shadyside, and East Hospitals intensive care units for assistance with coordination of patient enrollment and collection of patient samples. We would like to thank the laboratory personnel at the Center for Medicine and the Microbiome at the University of Pittsburgh for assistance with processing clinical samples. We acknowledge the contributions of Nameer Al-Yousif, MD, Michael Lu, MD, Grace Lisius MD, and Caitlin Shaefer, MPH who participated in clinical data extractions for specific components of the databases of the UPMC-ARF and UPMC-COVID cohorts. We also thank the Massachusetts General Hospital Translational and Clinical Research Center (TCRC) for their support of the project and the assembly of the MGH-COVID cohort.

## Author contributions

G.D.K.: Conceptualization, Methodology, Validation, Formal analysis, Investigation, Resources, Writing - Original Draft, Visualization, Supervision, Project administration, Funding acquisition; K.S.: Methodology, Validation, Investigation, Writing - Original Draft, Visualization; A.F., K.L., P.V.B., B.M., P.L.: Conceptualization, Methodology, Validation, Formal analysis, Investigation, Resources, Writing - Review & Editing, Visualization; H.Y., N.B., F.S., W.B., J.W.E., S.Q., X.W., A.P., Y.Z., J.R., C.D.C., D.A.O., C.Y.H., D.V.T.: Methodology, Investigation, Resources, Writing - Review & Editing; A.M., B.J.M.: Conceptualization, Methodology, Validation, Investigation, Resources, Writing - Review & Editing, Supervision, Project administration, Funding acquisition.

## Funding

Funding information Dr. Kitsios: University of Pittsburgh Clinical and Translational Science Institute, COVID-19 Pilot Award; NIH (R03 HL162655); American Lung Association COVID-19 Respiratory Virus Research Award. Dr. Benos: NIH (R01 HL157879; R01 HL127349, R01DK130294); Dr. McVerry: NIH (P01 HL114453); Dr. Bain: Veterans Affairs (IK2BX004886); Dr. Lai: Massachusetts General Hospital Translational and Clinical Research Center, supported by Grant 1UL1TR002541.

## Competing interests

Dr. Kitsios has received research funding from Karius, Inc, Genentech, Inc, and Pfizer, Inc, all unrelated to this project. Dr. Morris has received research funding from Pfizer, Inc., unrelated to this project. Dr. McVerry has received consulting fees from Boehringer Ingelheim, BioAegis, and Synairgen Research, Ltd. unrelated to this work. All other authors disclosed no conflict of interest.

## Ethics approval and consent to participate

The University of Pittsburgh Institutional Review Board (IRB) approved the protocol for the UPMC-ARF and UPMC-COVID cohorts (STUDY19050099). We obtained written or electronic informed consent by all participants or their surrogates in accordance with the Declaration of Helsinki. For the MGH-COVID cohort, the Study protocol #2020P000804 was approved by the Mass General Brigham IRB. For the healthy controls, the University of Pittsburgh IRB approved the study protocols (STUDY19060243 for respiratory biospecimens and STUDY20060312 for stool biospecimens). All participants or their healthcare proxy provided written informed consent to participate.

## Consent for publication

We obtained necessary patient/participant consent and the appropriate institutional forms have been archived. Any patient/participant/sample identifiers included were not known to anyone outside the research group so cannot be used to identify individuals.

## Additional information

[1]Division of Pulmonary, Allergy, Critical Care and Sleep Medicine, University of Pittsburgh, Pittsburgh, PA, USA. [2]Center for Medicine and the Microbiome, University of Pittsburgh, Pittsburgh, PA, USA. [3]Department of Epidemiology, University of Florida, Gainesville, FL, USA. [4]Department of Electrical and Computer Engineering & Computer Science, University of New Haven, West Haven, CT, USA. [5]School of Medicine, Tsinghua University, Beijing, China. [6]Division of Pulmonary Critical Care Medicine, Department of Medicine, Johns Hopkins University School of Medicine, Baltimore, MA, USA. [7]Veteran's Affairs Pittsburgh Healthcare System, Pittsburgh, PA, USA. [8]Division of Pulmonary and Critical Care Medicine, Massachusetts General Hospital and Harvard Medical School, Boston, MA, USA. [9]Division of Infectious Diseases, Department of Medicine, University of Pittsburgh, Pittsburgh, PA, USA. [10]These authors jointly supervised this work: Alison Morris, Bryan J. McVerry. ✉e-mail: kitsiosg@upmc.edu

