## [Peer Review File · Nature Communications]

REVIEWER COMMENTS

Reviewer #1 (Remarks to the Author):

The paper reports the occurrence of the progressive dysbiosis, characterized by reduced alpha diversity, depletion of obligate anaerobe bacteria, and pathogen enrichment, occurring in the oral, lung, and gut compartments in a large cohort of mechanically ventilated patients with acute respiratory failure. Despite the fact that chronic obstructive pulmonary disease, immunosuppression, and antibiotic exposure shaped dysbiosis, the unsupervised clusters of lung microbiota diversity and composition independently predicted survival, transcending clinical predictors. The authors conclude that insights into the dynamics of the microbiome during critical illness highlight the potential for microbiota-targeted interventions in precision medicine.

Comments

The paper is of high quality but many of the reported analyses lack an adequate comparison method since microbiome data obtained with high throughput sequencing are intrinsically compositional. The type and the quality of sequencing together with data normalization applied to each type of sequencing should be better detailed. The use of the limma package with clr normalization is highly appreciated but more details and/or source code for the regression model and at least a detailed workflow of your “custom Mothur-based pipeline” are appreciated. Attention should also be paid, in my opinion, to the method of estimating the Shannon index of each sample and its use. The Shannon index appears to be evaluated without applying any data normalization strategy and using all available readings in each sample. Therefore, the index evaluated with this procedure reflects the richness and uniformity of the microbial composition influenced by the sequencing depth of the sample. The use of the estimated index as described above, although it may be adequate for comparing groups obtained by DMM clustering (i.e. in Figure 3), could be misleading if used to compare diversity between groups as in Figure 1. In order to properly compare microbial diversity between groups, a data normalization strategy that takes sequencing depth into account should be considered, and the Shannon index of each sample should be estimated on the normalized data. This reviewer suggests using diversity index estimation procedures based on the intended purpose, i.e., comparing the compositional diversity or comparing clusters obtained considering all reads in each sample. Also the database and corresponding version on which taxonomy was evaluated should be explicitly indicated (supposed to be greengenes from some reference but please detail information) together to the classifier used.

Specific points

Some point of the “Supplementary Table 1 - The STORMS checklist” should be revised as requested below and text should be congruently adapted.

Line 56-67 – Abstract: Sequencing methods that define the strategy used for metagenomic classification are not indicated in the abstract while the STORMS checklist 1.2 state “yes”. Please update one of the two documents. If sequencing is inserted in the abstract, please specify the two types of sequencing: High-throughput (Illumina) and metagenome (nanopore).

STORM check list 3.6 - page 21: samples dropout should be further detailed. In particular, any dropout of the sequenced samples should be described. Please, specify exclusions due to eventual low number of reads as required by <https://www.stormsmicrobiome.org/figures/>.

Line 137: “In an initial quality control step, we demonstrated robust detection of bacterial 16S reads in oral, lung and gut samples in the UPMC-ARF cohort compared to negative controls (Figure S1A-B).”

Line 516: “For microbial community profiling, we included samples that produced >300 high quality microbial reads for both 16S-Seq and Nanopore sequencing.”

Figure S1: “A-B: Clinical samples from critically ill patients and healthy controls he had much higher sequencing yield (high quality reads from Illumina MiSeq 16S rRNA gene sequencing) and markedly different bacterial composition (as shown in Principal Coordinates Analysis) compared to negative controls or PCR positive samples.”

The text and figures caption should clearly describe which types of sequencing were used and if nanopore sequencing results are also included in the figure.

Figure S1A shows boxplots of the number of “high quality” reads from 16S Miseq Illumina. Being sequencing from high throughput compositional, the number of obtained reads, is “irrelevant, and contains only information on the precision of the estimate” (Gloor et al. 2017 <https://doi.org/10.3389/fmicb.2017.02224>). To make a robust comparison between groups, further information on the quality of 16S sequencing is needed since a low number of reads could be due to an inadequate sequencing depth even for samples with high microbial load. On the other hand, a high number of reads could be due to high PCR sample amplification even for samples with low microbial load. In particular, this reviewer suggests replacing the boxplots of the number of reads with the Shannon index rarefaction curves (at sequence variant or OTU level) of both clinical and control samples in order to assess whether a sequencing depth of 300 is enough to reach the plateau for all samples. In such a case, estimates of feature proportions will produce reliable Shannon estimate also for those samples with only 300 reads. Otherwise, a criterion will be available for removing samples that does not have enough reads for assuring a reliable feature proportion estimates. Also a Beta rarefaction analysis as reported by Cameron et al. 2021 at <https://doi.org/10.1038/s41598-021-01636-1> could help to evaluate if samples with small sequencing depth affects PCoA samples ordering.

Please, specify the metric on which PCoA is evaluated and highlight which normalization has been applied to the data (see Lin 2020 in <https://doi.org/10.1038/s41522-020-00160-w>). In fact, if PCoA is applied to diversity metrics evaluated on the taxa abundance in each sample without applying further normalization, the compositional diversity is heavily influenced by the different sequencing

depth between samples. Furthermore, describe the p value indicated in Figure S1 B (and D) since it is unclear which hypothesis it refers to. Also, the correction of p value should be defined since multiple comparisons are evaluated.

Please, specify the taxonomic level at which diversity indexes are evaluated (genus or sequence variants or OTU or something else).

In the method section some details should be given on the definition of “high quality reads” (i.e. number of average erroneous base per strand or per amplicon, minimum quality score per base,).

Line 143: “Samples from critically ill patients had significantly lower alpha diversity (Shannon index) in each compartment compared to corresponding healthy control samples. Alpha diversity further declined in all three body compartments across longitudinal samples (Figure 1A). Similarly, baseline ICU samples had markedly significant differences in beta diversity from healthy controls (Figure 1B)”

Line 516: “For microbial community profiling, we included samples that produced >300 high quality microbial reads for both 16S-Seq and Nanopore sequencing. We performed alpha diversity (Shannon index) calculations for each available sample, and then conducted between group comparisons of alpha diversity with non-parametric tests to draw inferences on systematic differences of alpha diversity between groups as a measure of relative community fitness. We conducted beta diversity analyses (Manhattan distances, analyzed via permutation analysis of variance and visualized via principal coordinates analyses) with the R vegan and mia packages.”

In order to properly compare alpha and beta indexes among groups, the type of normalization data, if any, should be described for each type of sequencing; whether all the data have an adequate number of reads that allow a reliable estimate of microbial composition should also be given, since the Illumina 16S sequencing is a non-quantitative method. As noted above, rarefaction curves of the Shannon index on each sample vs sequencing depth help in clarifying. Please give more details on Figure 1B in the text or at least in the figure caption: what metric the figure refers to and if the Permanova p values refers to the test on diversity values or on their corresponding PCoA. Also correction of p value should be defined since multiple comparisons are evaluated.

Line 146: “Taxonomic composition comparisons showed depletion of multiple commensal taxa in ICU samples, with significant enrichment for Staphylococcus in oral and lung samples, and Anaerococcus and Staphylococcus in gut samples (Figure 1C-D-E).”

Please specify the two groups between which the logFC is evaluated. It can be argued from Figure 1-A and labels in Figure 1-D that the two groups are all ICU patients and the healthy control but it seems it is not specified in the text or in the Figure legend.

Line 151: “We then examined the compositional similarity (Bray-Curtis indices) between compartments to understand the relationship between the low biomass (lung) vs. high biomass (oral and gut) communities. We found higher similarity between oral-lung vs. gut-lung communities in the baseline and middle intervals (Figure 1G).”

Figure 1G and its description are confusing. For each point time the figure axis reports the BC dissimilarity while in the text is described the BC similarity. Please clarify.

Assuming the correctness of the analysis shown in the figure, a BC dissimilarity index close to one indicates a high degree of diversity between the two groups in terms of quantitative composition. As such, the figure indicates that, for each time point, BC diversity between lung and gut is really high, close to 1, while BC diversity between oral and lung is lower. Authors should specify and describe the reason of the comparison of the average between oral-lung and gut-lung diversities or may be use the BC diversity of oral, lung and gut separately and mutually compare them. For a straightforward interpretation of the BC dissimilarity, the explanation of the data were normalized is recommended.

Line 154: "Taxonomic comparisons between compartments revealed that no specific taxa were systematically different between oral and lung microbiota (Figure 1H),"

The sentence should be reformulated since figure shows many feature above the significance threshold equal although with a corresponding logFC lower than 1.5.

Data for differential abundance analysis in limma have been normalized with log-transform methods (clr). Please indicate whether rarefaction was also applied or whether the prevalence, in addition to the FC threshold, of feature was considered to significantly associate a feature to a group. Prevalence could help also in clarifying data reported in Figure S2.

Line 170: "In both oral and lung communities, we found a progressive decline in the relative 171 abundance of obligate anaerobes over time. There was, however, no corresponding change in the gut 172 composition of anaerobic (obligate or facultative) bacteria over time (Figure 2A-B)."

The comparisons of feature relative abundance among groups also need to be carried out upon the explicit definition of the data normalization procedure. Being high-throughput sequenced data compositional in nature, traditional statistical tests cannot be applied to feature proportions unless properly normalized. This reviewer suggests to use clr as the normalization procedure for features relative abundance, as the authors have already done in Figure 1 C-D-E-H-I and as suggested by Gloor 2017 in the reference cited above. The same comment applies to analyses described in Figure S2 and 3.

Line 496: "We processed derived 16S sequences with a custom Mothur-based pipeline and performed analyses at genus level." Details of the custom pipeline should be given by describing the bioinformatics workflow and/or making available the pipeline code.

Line 570: "To develop these models in each compartment (oral, lung and gut), we used probabilistic graphical modeling (PGM) by considering the 50 most abundant taxa in each compartment along with the Shannon Index." Please, indicate explicitly what type of abundance was used (absolute, relative, clr, ...) in the MLR model equation.

Line 662: PRJNA726955 is not available on NCBI. Please check.

Line 666: "Primary code and de-identified data for replication of analyses will be available on the github repository (<https://github.com/MicrobiomeALIR/MultiCompartmentMicrobiome>) upon acceptable of the manuscript for Publication." A confidential link to the reviewer for review purposes would be greatly appreciated and would speed up the review process.

On a final note, it would be quite helpful if the authors precisely define which genera are included in the classification of bacterial genera into true respiratory pathogens, or oral commensals, or “other” category. For instance, what is the biological meaning of having plausible respiratory pathogens expanded in the gut at the baseline? Who were those? Likewise, how the authors interpret the abundance of the gut commensal *C. albicans* in the lung? Is it a respiratory pathogen? If so, how? Obligate anaerobes progressively declined in both oral and lung communities but not in the gut. Please, indicate here some of the relevant genera (among obligate anaerobes, facultative anaerobes, aerobes and microaerophiles), it will facilitate the comprehension. Likewise, specify the anaerobe abundance in patients with COPD and what the increased pathogen abundance refers to in all three compartments. The fact that receipt of anaerobic spectrum antibiotics was associated with a progressive decrease in obligate anaerobe abundance, without significant effects on pathogen abundance, does it mean that facultative bacteria were spared? How do you explain the the gut-origin taxa enrichment in the lung samples without overt oropharyngeal colonization with such taxa?

Reviewer #2 (Remarks to the Author):

Kitsios et al. present a longitudinal multicenter cohort study investigating dynamic changes in the oral, lung, and gut microbiota in hospitalized critically ill patients and the impact of these changes on pertinent clinical outcomes. This is an important study that further advances our understanding of the prognostic relevance of microbiota signatures in critically ill patients.

The investigators astutely point out that most microbiota studies involving critically ill patients are limited by statistical models that incorporate only a single microbial signature at a fixed sampling timepoint.

They found that dysbiotic signatures emerged in all three body compartments (lung, gut, and oral cavity), predominantly characterized by a decrease in alpha diversity, a depletion of obligate anaerobe bacteria, and increased enrichment in more traditionally pathogenic taxa. In addition, they used DMM and bacterial-fungal SNF clustering methods to identify microbial signatures in the lungs (e.g., low diversity in the case of DMM clusters, high pathogen abundance in the case of bacterial-fungal SNF clusters) that were associated with worse survival.

Overall, this is a well-conducted study that addresses several novel questions. Strengths of this study include its large patient sample size (with a vast number of oral, LRT, and gut samples), its use of both derivation and validation cohorts, its use of longitudinally collected samples, and its use of methods that extend beyond 16S rRNA gene sequencing (which is limited solely to bacterial taxonomy). The manuscript contains a substantial amount of data but is concisely and clearly written.

Major comments:

1. The derivation cohort was more than 90% white/Caucasian, which may limit generalizability, although their findings were validated in more racially diverse cohorts (72-76% white). Nonetheless, this should be highlighted as a limitation.
2. I was surprised to see a lack of taxonomic differences between the oral and lung microbiota (Figure 1H). To clarify, was this looking collectively at all oral and LRT samples? Is it possible that differences might emerge if you stratify by sample collection timepoint? Do the investigators think there would have been differences if they were comparing oral samples to BAL samples rather than to tracheal aspirates?
3. Did the authors perform any type of contamination analysis? They very clearly demonstrate low biomass in their control samples. However, there is no designation of which individual ASVs/genera potentially represent contaminants. This is due mainly to the fact that microbes were clustered according to their oxygen requirement and pathogenicity (two classification schemes with clinical relevance in one prior study of patients with aspiration pneumonia) rather than genera.
4. Can the authors provide a figure demonstrating what happens to bacterial load in all three compartments over time?
5. Was there any consideration given to presenting these data with stratification according to genera rather than oxygen requirement and pathogenicity designations? What was the basis for these groupings? They only provide a single citation justifying this approach.
6. What were the major taxonomic differences between DMM and weighted SNF clusters? This should be more clearly stated.
7. In Figure 4, are the predictors used in the survival analyses DMM and SNF designations at any collection timepoint? Or does this only constitute the first sampling timepoint? If it is any sampling timepoint, the study loses its ability in part to look at how dynamic changes affect survival, which is really what drives home the novelty of this investigation. They point out that there was very little shift between DMM clusters based on the timing of sample collection. Then again, they also point out how the number of samples dwindled in later sample collection timepoints. Nonetheless, these analyses could be modeled using joint modeling of longitudinal and time-to-event data (see Henderson et al., Biostatistics 2000) with reporting of association strength estimates.
8. The authors clearly state that use of endotracheal aspirate samples to represent the lung microbiota as opposed to reference standard BAL is a limitation of their study. They should comment more on how their findings might have differed if they used BAL samples.
9. The authors point out a lack of association between inflammatory sub-phenotypes and DMM microbiota clusters. Regarding the lungs, multiple studies to date have shown that the lower airway microbiota affects lower airway immune tone in both diseased and healthy individuals. Do they investigators have any insight as to why no association was noted in their cohort? Is this because of their use of tracheal aspirates rather than BAL?

Minor comments:

1. A number of abbreviations are used in Figure S1 that are not clearly or accessibly defined.
2. As part of their supplement, the authors should specify which specific taxa/genera were placed into the oxygen requirement and pathogenicity categories.
3. The authors might consider regressing bacterial burden against the clinical covariates studied in other linear regression models.
4. How was “immunosuppression” defined in regression models?
5. I might consider adding grid lines to the images in Figure S4 to ease readability.

Georgios Kitsios, MD, PhD
Assistant Professor of Medicine

Pittsburgh, March 25, 2024

To Reviewers 1 and 2:

We are deeply thankful for your thorough assessment and insightful feedback on our manuscript. Your input has helped us to further improve the quality of our manuscript. In this major revision, we address all criticisms and provide a detailed point-by-point review to all criticisms, along with the corresponding edits made to the manuscript.

We would like to point out that we implemented all transformation and filtering analyses proposed by Reviewer 1, with provision of primary code and de-identified data which improved the clarity and interpretability of our findings. We also implemented all additional analyses recommended by Reviewer 2, including joint modeling for assessment of longitudinal microbiota changes and impact on patient outcomes. While our overall conclusions remain similar to our original submission, we have obtained further insights into our dataset, with important findings highlighted in the main manuscript and provision of extensive data and display items in the Supplement.

All authors have read and approved this submitted manuscript, which is not currently under consideration for publication in any other journal.

We hope that you will consider our manuscript of appropriate quality, rigor and importance for publication in *Nature Communications*.

Sincerely,

Georgios D. Kitsios, MD, PhD
Assistant Professor of Medicine
Division of Pulmonary, Allergy, Sleep and Critical Care Medicine
University of Pittsburgh Medical Center
Address: UPMC Montefiore Hospital, NW628,
3459 Fifth Avenue, Pittsburgh, PA 15213
Email: kitsiosg@upmc.edu

Responses to Reviewer Comments

Reviewer #1 (Remarks to the Author):

The paper reports the occurrence of the progressive dysbiosis, characterized by reduced alpha diversity, depletion of obligate anaerobe bacteria, and pathogen enrichment, occurring in the oral, lung, and gut compartments in a large cohort of mechanically ventilated patients with acute respiratory failure. Despite the fact that chronic obstructive pulmonary disease, immunosuppression, and antibiotic exposure shaped dysbiosis, the unsupervised clusters of lung microbiota diversity and composition independently predicted survival, transcending clinical predictors. The authors conclude that insights into the dynamics of the microbiome during critical illness highlight the potential for microbiota-targeted interventions in precision medicine.

Comments

The paper is of high quality but many of the reported analyses lack an adequate comparison method since microbiome data obtained with high throughput sequencing are intrinsically compositional. The type and the quality of sequencing together with data normalization applied to each type of sequencing should be better detailed. The use of the limma package with clr normalization is highly appreciated but more details and/or source code for the regression model and at least a detailed workflow of your “custom Mothur-based pipeline” are appreciated. Attention should also be paid, in my opinion, to the method of estimating the Shannon index of each sample and its use. The Shannon index appears to be evaluated without applying any data normalization strategy and using all available readings in each sample. Therefore, the index evaluated with this procedure reflects the richness and uniformity of the microbial composition influenced by the sequencing depth of the sample. The use of the estimated index as described above, although it may be adequate for comparing groups obtained by DMM clustering (i.e. in Figure 3), could be misleading if used to compare diversity between groups as in Figure 1. In order to properly compare microbial diversity between groups, a data normalization strategy that takes sequencing depth into account should be considered, and the Shannon index of each sample should be estimated on the normalized data. This reviewer suggests using diversity index estimation procedures based on the intended purpose, i.e., comparing the compositional diversity or comparing clusters obtained considering all reads in each sample. Also the database and corresponding version on which taxonomy was evaluated should be explicitly indicated (supposed to be greengenes from some reference but please detail information) together to the classifier used.

Response: Thank you for this very positive assessment and thorough review of our manuscript. In this revision, we provide further details with regards to all analytical steps taken for generating our findings, and address all your suggestions below in full detail.

With regards to the custom Mothur-based pipeline, we provide the link to the github source code, beyond citations to previous work from our group utilizing this workflow.

“We processed derived 16S sequences with a custom Mothur-based pipeline (v1.44.1) as previously described (available at https://github.com/MedicineAndTheMicrobiome/AnalysisTools/tree/master/16S_Clust_Gen_Pipeline).”

We provide detailed responses for the specific comments on Shannon diversity estimation and data normalization strategies in our responses that follow.

Per the Reviewer’s strong recommendations, we have applied rarefaction prior to diversity analyses (alpha and beta), and provide all details in our responses below.

We specify the rarefaction parameters and results with the following sections in Methods, Results and new Figure S1:

“We considered clinical samples that generated $\geq 1,000$ quality 16S-Seq reads and performed rarefaction at 1,000 reads to control for uneven sequencing depth between samples in estimation of diversity indices: alpha diversity by Shannon index (Figure S1), and beta diversity by Bray-Curtis similarity index in centered-log ratio (CLR) transformed abundances.”

“We calculated alpha diversity with the Shannon index following rarefaction at 1,000 reads, with 100 random subsamplings to obtain average Shannon index for each available sample. Rarefaction curves showed that all clinical samples had reached plateau for calculation of Shannon index by 1,000 reads (Figure S1). We applied rarefaction to allow for between sample type comparisons because sequencing depth and yield varied between sample types.”

With regards to the taxonomy database and classifier used, we clarify the specifics in our revised Online Methods section:

For 16S-Seq:

“Sequence taxonomic classifications were performed with the Ribosomal Database Project’s (RDP) naïve Bayesian classifier with the SILVA 16S rRNA database (v138).”

For ITS-Seq:

“Chimeras were removed and the Unite database was utilized to classify reads into amplicon sequence variants (ASVs) using the naïve Bayesian classifier method, defined at the species level.”

For Nanopore:

“We analyzed microbial metagenomic sequences with the EPI2ME platform (ONT) and the “What’s In My Pot” [WIMP] workflow to quantify abundance of microbial species.”

For Illumina NovaSeq metagenomics sequencing (MGH validation cohort only)

“Taxonomic profiles were generated using the bioBakery 3 shotgun metagenome workflow 3.0.0, the details of which have previously been described. Briefly, human reads were filtered using KneadData 0.10.0 and species-level taxonomic profiles generated using MetaPhlan 3.0.0.”

Specific points

R1C1: Some point of the “Supplementary Table 1 - The STORMS checklist” should be revised as requested below and text should be congruently adapted.

Response: Thank you, we revised the STORMS checklist accordingly.

R1C2: Line 56-67 – Abstract: Sequencing methods that define the strategy used for metagenomic classification are not indicated in the abstract while the STORMS checklist 1.2 state “yes”. Please update one of the two documents. If sequencing is inserted in the abstract, please specify the two types of sequencing: High-throughput (Illumina) and metagenome (nanopore).

Response: Thank you for pointing this out. We added the following sentence in the Abstract to illustrate the use of different sequencing techniques in our study.

“Employing advanced DNA sequencing technologies, including Illumina amplicon sequencing (utilizing 16S and ITS rRNA genes for bacteria and fungi, respectively, in all sample types) and Nanopore metagenomics for lung microbiota, we observed progressive dysbiosis in all three body compartments.”

R1C3: STORM check list 3.6 - page 21: samples dropout should be further detailed. In particular, any dropout of the sequenced samples should be described. Please, specify exclusions due to eventual low number of reads as required by <https://www.stormsmicrobiome.org/figures/>.

Response: We provide details on the sample dropout in the revised Methods section:

“In the UPMC-ARF cohort, we considered 1520 unique clinical samples (593 oral swabs, 578 ETA/BALF [lung], and 349 stool or soiled rectal swabs [gut]). Filtering at 1,000 reads resulted in elimination of 112 clinical samples (24 oral swabs, 77 lung samples, and 11 gut samples). We performed 16S-Seq analyses at the genus level, which were filtered for singletons and low abundance taxa (i.e. those with relative abundance <0.0001 in <5% of samples), resulting in a final set of 214 unique genera for analyses.”

R1C4: Line 137: “In an initial quality control step, we demonstrated robust detection of bacterial 16S reads in oral, lung and gut samples in the UPMC-ARF cohort compared to negative controls (Figure S1A-B).”

Line 516: “For microbial community profiling, we included samples that produced >300 high quality microbial reads for both 16S-Seq and Nanopore sequencing.”

Figure S1: “A-B: Clinical samples from critically ill patients and healthy controls he had much higher sequencing yield (high quality reads from Illumina MiSeq 16S rRNA gene sequencing) and markedly different bacterial composition (as shown in Principal Coordinates Analysis) compared to negative controls or PCR positive samples.”

The text and figures caption should clearly describe which types of sequencing were used and if nanopore sequencing results are also included in the figure.

Response: We apologize for any confusion in which sequencing approach was used for our analyses. We now clarify which methodology (16S-Seq; ITS-Seq; Nanopore Metagenomics) was used in each section of the results.

In the revised Results, we added the following sentences:

“By Illumina MiSeq 16S-Seq, we analyzed a total of 2557 clinical samples among all three cohorts and healthy controls, as well as 233 experimental control samples obtained either during patient sampling at the bedside or during sample processing in the laboratory.”

*“Beyond 16S-Seq, Illumina MiSeq Fungal ITS-Seq showed that >50% of communities in all three compartments were dominated by *C. albicans* (defined as >50% relative abundance), with a progressive decline in fungal Shannon index in oral and lung communities during follow-up. Nanopore DNA metagenomics of lung samples provided similar bacterial representations to 16S-Seq analyses and confirmed high abundance of *C.albicans* detected by ITS-Seq.”*

All Figure legends explicitly state the sequencing methodology used to generate the displayed results.

R1C5: Figure S1A shows boxplots of the number of “high quality” reads from 16S Miseq Illumina. Being sequencing from high throughput compositional, the number of obtained reads, is “irrelevant, and contains only information on the precision of the estimate” (Gloor et al.

2017 <https://doi.org/10.3389/fmicb.2017.02224>). To make a robust comparison between groups, further information on the quality of 16S sequencing is needed since a low number of reads could be due to an inadequate sequencing depth even for samples with high microbial load. On the other hand, a high number of reads could be due to high PCR sample amplification even for samples with low microbial load. In particular, this reviewer suggests replacing the boxplots of the number of reads with the Shannon index rarefaction curves (at sequence variant or OTU level) of both clinical and control samples in order to assess whether a sequencing depth of 300 is enough to reach the plateau for all samples. In such a case, estimates of feature proportions will produce reliable Shannon estimate also for those samples with only 300 reads. Otherwise, a criterion will be available for removing samples that does not have enough reads for assuring a reliable feature proportion estimates. Also a Beta rarefaction analysis as reported by Cameron et al. 2021 at <https://doi.org/10.1038/s41598-021-01636-1> could help to evaluate if samples with small sequencing depth affects PcoA samples ordering.

Response: Thank you for this very helpful comment. In response to your suggestions, we provide the detailed rarefaction curves that illustrate the selection of a minimum threshold of 1,000 reads for the clinical samples analyzed. We provide a new supplemental Figure with all the details on rarefaction and PCoA plotting.

“Figure S1: Quality control steps for clinical and experimental control samples by Illumina MiSeq 16S rRNA gene sequencing. A. Clinical samples from ICU patients and healthy control subjects had much higher 16S-Seq yield of high-quality reads compared to experimental and procedure negative controls. Red dashed line at 1,000 reads. B. Shannon index (alpha diversity) rarefaction curves by sequencing depth for the clinical samples from ICU patients (oral, lung, and gut), illustrating that all samples reached plateau of Shannon indices at the cut-off of 1,000 reads used in analyses. C. Clinical samples had markedly different bacterial composition based on Centered-Log-Ratio (CLR)-transformed abundances of bacterial genera, shown as Bray-Curtis indices in Principal Coordinates Analysis (PCoA) compared to negative controls or PCR positive samples (Zymo Mock community controls), and analyzed by permutational analysis of variance (Permanova). D. Top taxa detected in negative control samples, indicating possible contamination. Overall, such taxa were detected at very low levels (mean number of reads <100). E-F. Quality control examination for gut samples. Unsoiled rectal swabs (i.e. not visibly coated by stool) had markedly lower bacterial burden (examined by qPCR of 16S rRNA gene) and differential composition compared to soiled rectal swabs or stool samples, and therefore, unsoiled rectal swabs were excluded from further analysis as they may not provide sufficient representation of gut microbiota.

.”

We also added the following text in Methods to justify the use of rarefaction for diversity parameter estimation, as per the Reviewer’s suggestion, and provide citations to the papers suggested by the Reviewer, as well as to the recent publication by Patrick Schloss on the advantages of rarefaction (PMID: 38054712).

“We calculated alpha diversity with the Shannon index following rarefaction at 1,000 reads, with 100 random subsamplings to obtain average Shannon index for each available sample. Rarefaction curves showed that all clinical samples had reached plateau for calculation of Shannon index by 1,000 reads (Figure S1). We applied rarefaction to allow for between sample type comparisons because sequencing depth and yield varied between sample types. We conducted between group comparisons of alpha diversity with non-parametric tests to draw inferences on systematic differences of alpha diversity between groups as a measure of relative community fitness.¹ We conducted beta diversity analyses (Bray-Curtis indices) on centered log-ratio (CLR) transformed abundances, analyzed via permutation analysis of variance (Permanova) and visualized via principal coordinates analyses with the R vegan and mia packages.”

R1C6: Please, specify the metric on which PCoA is evaluated and highlight which normalization has been applied to the data (see Lin 2020 in <https://doi.org/10.1038/s41522-020-00160-w>). In fact, if PCoA is applied to diversity metrics evaluated on the taxa abundance in each sample without applying further normalization, the compositional diversity is heavily influenced by the different sequencing depth between samples. Furthermore, describe the p value indicated in Figure S1 B (and D) since it is unclear which hypothesis it refers to. Also, the correction of p value should be defined since multiple comparisons are evaluated.

Response: Thank you. We clarify that we performed PCoA plots on Bray-Curtis indices from CLR transformed abundances. The p-values are from Permanova tests (*adonis* function). We report globally adjusted p-value for the number of Permanova tests performed and reported per the Bonferroni corrected method.

We added the following text in the Methods section:

“We conducted beta diversity analyses (Bray-Curtis indices) on centered log-ratio (CLR) transformed abundances, analyzed via permutation analysis of variance (Permanova) and visualized via principal coordinates analyses with the R vegan and mia packages. We adjusted all reported p-values from Permanova tests for multiple testing with a conservative Bonferroni correction.”

R1C7: Please, specify the taxonomic level at which diversity indexes are evaluated (genus or sequence variants or OTU or something else).

Response: We performed all 16S analyses at genus level, whereas for ITS-Seq, Illumina NovaSeq metagenomics and Nanopore metagenomics we analyzed data at species level.

We added the following text in the Methods section:

“We performed 16S-Seq analyses at the genus level, which were filtered for singletons and low abundance taxa (i.e. those with relative abundance <0.0001 in <5% of samples), resulting in a final set of 214 unique genera for analyses. We analyzed ITS-Seq at species level for fungi, and Nanopore metagenomics at species level for DNA reads of bacteria, fungi and viruses.”

R1C8: In the method section some details should be given on the definition of “high quality reads” (i.e. number of average erroneous base per strand or per amplicon, minimum quality score per base,).

Response: We added the following text in the Methods section for the definition of quality reads for each sequencing technology.

For 16S-Seq:

“We processed derived 16S sequences with a custom Mothur-based pipeline (v1.44.1) as previously described (available at https://github.com/MedicineAndTheMicrobiome/AnalysisTools/tree/master/16S_Clust_Gen_Pipeline). In brief, we deconvoluted sequences from the Illumina MiSeq and processed them through an in-house sequence quality control pipeline, which includes dust low complexity filtering, quality value (QV<25) trimming, and trimming of primers used for 16S rRNA gene amplification, and minimum read length filtering. Trimmed reads shorter than 75bp or those with less than 95% of the bases above a QV of 25 were discarded. Forward and reversed paired reads were merged into contigs, and processed for 16S rRNA gene sequence clustering and annotation pipeline.”

For Nanopore Metagenomics:

"We analyzed microbial metagenomic sequences with the EPI2ME platform (ONT) and the "What's In My Pot" [WIMP] workflow to quantify abundance of microbial species.⁴⁵ We filtered FASTQ files with a mean quality (q-score) below a minimum threshold of 7."

R1C9: Line 143: "Samples from critically ill patients had significantly lower alpha diversity (Shannon index) in each compartment compared to corresponding healthy control samples. Alpha diversity further declined in all three body compartments across longitudinal samples (Figure 1A). Similarly, baseline ICU samples had markedly significant differences in beta diversity from healthy controls (Figure 1B)"

Line 516: "For microbial community profiling, we included samples that produced >300 high quality microbial reads for both 16S-Seq and Nanopore sequencing. We performed alpha diversity (Shannon index) calculations for each available sample, and then conducted between group comparisons of alpha diversity with non-parametric tests to draw inferences on systematic differences of alpha diversity between groups as a measure of relative community fitness. We conducted beta diversity analyses (Manhattan distances, analyzed via permutation analysis of variance and visualized via principal coordinates analyses) with the R vegan and mia packages."

In order to properly compare alpha and beta indexes among groups, the type of normalization data, if any, should be described for each type of sequencing; whether all the data have an adequate number of reads that allow a reliable estimate of microbial composition should also be given, since the Illumina 16S sequencing is a non-quantitative method. As noted above, rarefaction curves of the Shannon index on each sample vs sequencing depth help in clarifying. Please give more details on Figure 1B in the text or at least in the figure caption: what metric the figure refers to and if the Permanova p values refers to the test on diversity values or on their corresponding PCoA. Also correction of p value should be defined since multiple comparisons are evaluated.

Response: Thank you. Per your suggestion, we have now implemented rarefaction prior to diversity calculations.

We also provide citation to recent literature in the field supporting the use of rarefaction for diversity index calculations.

We have revised Figure 1 and now present the diversity indices that were obtained following rarefaction, as detailed in our revised Methods.

We provide a detailed caption for Figure 1 that explains which metrics are presented in each panel and which comparison they represent. To facilitate interpretation, we have broken Figure 1 in a top set of panels (A-C) that provides intra-compartment comparisons between ICU patients and healthy controls, and a bottom set of panels (D-F) with inter-compartment comparisons among ICU patients.

"Figure 1. Intra- and inter-compartment comparisons of microbiota profiles by Illumina 16S-Seq reveal features of dysbiosis in all three body compartments in critically ill patients. Panels A-C: Intra-compartment comparisons between ICU patients and healthy controls. A. Samples from critically ill patients had significantly lower alpha diversity (Shannon index obtained post-rarefaction with random subsampling of reads in samples with $\geq 1,000$ 16S rRNA gene reads) compared to corresponding healthy control samples in each compartment ($p < 0.001$), with further decline of Shannon index over time in longitudinal samples in critically ill patients ($p < 0.001$). B. Baseline samples from critically ill patients had markedly significant differences in beta diversity (Bray-Curtis indices in centered-log ratio transformed [CLR] abundances following random subsampling of reads in samples with $\geq 1,000$ reads) compared to healthy controls (visualized with Principal Coordinates Analysis [PCoA] and statistically compared with permutational analysis of variance [permanova] p-values < 0.001 , adjusted for multiple comparisons with the Bonferroni method). C. Taxonomic composition comparisons with the limma package showed high effect sizes and significance thresholds (threshold of \log_2 -fold-change [\log_2FC] of CLR-transformed abundances > 1.5 ; Benjamini-Hochberg adjusted p-value < 0.05) showed depletion for multiple commensal taxa in critically ill patients samples, with significant enrichment for *Staphylococcus* in oral and lung samples, and *Anaerococcus* and *Staphylococcus* in gut samples (significant taxa shown in red in the volcano plots). Panels D-F: Inter-

compartment comparisons among ICU patients. D: Lung samples had lower bacterial burden compared to oral and gut samples by 16S rRNA gene qPCR (all $p < 0.001$). E. PCoA plot of beta-diversity shows compositional similarity for the oral and lung compartments, which were compositionally dissimilar to gut samples (permanova $p < 0.001$). F. Taxonomic comparisons between compartments revealed that no specific taxa were systematically different between oral and lung microbiota above the threshold of $\log_{2}FC \geq 1.5$, whereas in gut-lung comparisons, lung communities were enriched for typical respiratory commensals (e.g. *Rothia*, *Veillonella*, *Streptococcus*) and gut communities for gut commensals (e.g. *Bacteroides*, *Lachnospiraceae*, *Lachnospiraceae*).”

We also edited the corresponding section of Results accordingly:

“Following these quality-control steps, we first performed intra-compartment comparisons of samples from critically ill patients from the UPMC-ARF cohort to healthy control samples. At baseline, critically-ill patients had significantly lower alpha diversity in each compartment compared to corresponding healthy control samples. Despite the low Shannon index at baseline for ICU patients, their Shannon index further declined in all three body compartments in longitudinal samples (Figure 1A). Similarly, baseline ICU samples had markedly significant differences in beta diversity from healthy controls (Figure 1B). By taxonomic comparisons of CLR-transformed abundances within each compartment at baseline, ICU patient samples showed depletion

of multiple commensal taxa, with significant enrichment for *Staphylococcus* in oral and lung samples, and *Anaerococcus* and *Staphylococcus* in gut samples (Figure 1C).

We then performed inter-compartment comparisons among ICU samples. Bacterial load quantification by 16S qPCR confirmed that the LRT (lungs) had significantly lower biomass compared to URT (oral) and gastrointestinal tract (stool or soiled rectal swabs, Figure 1D). By beta-diversity comparisons (Bray-Curtis indices), oral and lung communities had high compositional similarity, whereas gut samples were compositionally different compared to oral and lung microbiota (Figure 1E). Taxonomic comparisons of CLR-transformed abundances between compartments revealed that no specific taxa were systematically different between oral and lung microbiota (Figure 1H), whereas in gut-lung comparisons, lung communities were enriched for typical respiratory commensals (e.g. *Rothia*, *Veillonella*, *Streptococcus*) and gut communities for gut commensals (e.g. *Bacteroides*, *Lachnospiraceae_uncl*) (Figure 1F).”

R1C10: Line 146: “Taxonomic composition comparisons showed depletion of multiple commensal taxa in ICU samples, with significant enrichment for *Staphylococcus* in oral and lung samples, and *Anaerococcus* and *Staphylococcus* in gut samples (Figure 1C-D-E).”

Please specify the two groups between which the logFC is evaluated. It can be argued from Figure 1-A and labels in Figure 1-D that the two groups are all ICU patients and the healthy control but it seems it is not specified in the text or in the Figure legend.

Response: Please also refer to our response to R1C9 above. We improved the clarity of presentation of Figure 1 with annotations of enrichment/depletion in each group, and now specify in the legend the groups being compared.

“C. Taxonomic composition comparisons with the *limma* package showed high effect sizes and significance thresholds (threshold of log₂-fold-change [logFC] of CLR-transformed abundances >1.5; Benjamini-Hochberg adjusted p-value<0.05), revealing depletion for multiple commensal taxa in critically ill patients samples, with significant enrichment for *Staphylococcus* in oral and lung samples, and *Anaerococcus* and *Staphylococcus* in gut samples (significant taxa shown in red in the volcano plots).”

R1C11: Line 151: “We then examined the compositional similarity (Bray-Curtis indices) between compartments to understand the relationship between the low biomass (lung) vs. high biomass (oral and gut) communities. We found higher similarity between oral-lung vs. gut-lung communities in the baseline and middle intervals (Figure 1G).”

Figure 1G and its description are confusing. For each point time the figure axis reports the BC dissimilarity while in the text is described the BC similarity. Please clarify.

Assuming the correctness of the analysis shown in the figure, a BC dissimilarity index close to one indicates a high degree of diversity between the two groups in terms of quantitative composition. As such, the figure indicates that, for each time point, BC diversity between lung and gut is really high, close to 1, while BC diversity between oral and lung is lower. Authors should specify and describe the reason of the comparison of the average between oral-lung and gut-lung diversities or may be use the BC diversity of oral, lung and gut separately and mutually compare them. For a straightforward interpretation of the BC dissimilarity, the explanation of the data were normalized is recommended.

Response: We understand that the previous depiction of BC indices between compartments did not have a straightforward interpretation. We adopted your suggestion and now demonstrate a PCoA plot and a permanova analysis with these inter-compartment comparisons among ICU patients. We clarify that the PCoA plots represent normalized data (post-rarefaction and CLR-transformed).

“B. Baseline samples from critically ill patients had markedly significant differences in beta diversity (Bray-Curtis indices in centered-log ratio transformed [CLR] abundances following random subsampling of reads in samples with ≥ 1,000 reads) compared to healthy controls (visualized with Principal Coordinates Analysis [PCoA] and statistically compared with permutational analysis of variance [permanova] p-values <0.001,

adjusted for multiple comparisons with the Bonferroni method). ... E. PCoA plot of beta-diversity shows compositional similarity for the oral and lung compartments, which were compositionally dissimilar to gut samples (permanova $p < 0.001$).

The corresponding section of Results now reads:

“We then performed inter-compartment comparisons among ICU samples. Bacterial load quantification by 16S qPCR confirmed that the LRT (lungs) had significantly lower biomass compared to URT (oral) and gastrointestinal tract (stool or soiled rectal swabs, Figure 1D). By beta-diversity comparisons (Bray-Curtis indices), oral and lung communities had high compositional similarity, whereas gut samples were compositionally different compared to oral and lung microbiota (Figure 1E).”

R1C12: Line 154: “Taxonomic comparisons between compartments revealed that no specific taxa were systematically different between oral and lung microbiota (Figure 1H),”

The sentence should be reformulated since figure shows many feature above the significance threshold equal although with a corresponding logFC lower than 1.5.

Data for differential abundance analysis in limma have been normalized with log-transform methods (clr). Please indicate whether rarefaction was also applied or whether the prevalence, in addition to the FC threshold, of feature was considered to significantly associate a feature to a group. Prevalence could help also in clarifying data reported in Figure S2.

Response: We modified the legend of Figure 1 to the following to clarify that no differentially abundant taxa exceeded the threshold of logFC 1.5, hence none were annotated in the oral vs. lung comparison.

“C. Taxonomic composition comparisons with the limma package showed high effect sizes and significance thresholds (threshold of log2-fold-change [logFC] of CLR-transformed abundances > 1.5 ; Benjamini-Hochberg adjusted p -value < 0.05) showed depletion for multiple commensal taxa in critically ill patients samples, with significant enrichment for Staphylococcus in oral and lung samples, and Anaerococcus and Staphylococcus in gut samples (significant taxa shown in red in the volcano plots) ... F. Taxonomic comparisons between compartments revealed that no specific taxa were systematically different between oral and lung microbiota above the threshold of $\log_{2}FC \geq 1.5$, whereas in gut-lung comparisons, lung communities were enriched for typical respiratory commensals (e.g. Rothia, Veillonella, Streptococcus) and gut communities for gut commensals (e.g. Bacteroides, Lachnoclostridium, Lachnospiraceae).”

We did not perform any differential abundance comparisons in former Figure S2 (analysis for enrichment of oral and lung communities with gut-origin bacteria), therefore we did not apply any logFC thresholds.

R1C13: Line 170: “In both oral and lung communities, we found a progressive decline in the relative 171 abundance of obligate anaerobes over time. There was, however, no corresponding change in the gut 172 composition of anaerobic (obligate or facultative) bacteria over time (Figure 2A-B).”

The comparisons of feature relative abundance among groups also need to be carried out upon the explicit definition of the data normalization procedure. Being high-throughput sequenced data compositional in nature, traditional statistical tests cannot be applied to feature proportions unless properly normalized. This reviewer suggests to use clr as the normalization procedure for features relative abundance, as the authors have already done in Figure 1 C-D-E-H-I and as suggested by Gloor 2017 in the reference cited above. The same comment applies to analyses described in Figure S2 and 3.

Response: This is a very helpful suggestion. We revised our analyses for the abundance of different categories of bacterial taxa (by oxygen requirement or plausible pathogenicity) and conducted pairwise comparisons on CLR transformed abundances. As also per the suggestion of R1C6, we adjusted our p -values with the Bonferroni method for multiple comparisons.

Please note that for display purposes and interpretability of abundance taxa, we maintained the representation of Figure 2 as relative abundance barplot and boxplots. However, we explain clearly in the Figure legend that the p-values represent comparisons of CLR-transformed abundances and that they are adjusted for multiple comparisons.

The revised Figure 2 legend now reads:

“Figure 2: Longitudinal analysis of bacterial composition showed a progressive loss of obligate anaerobes in oral and lung communities as well as enrichment for recognized respiratory pathogens in all three compartments. Top Panels (A-B): Relative abundance barplots for oral, lung and gut samples with classification of bacterial genera by oxygen requirement into obligate anaerobes (anaerobes), aerobes, facultative anaerobes, microaerophiles, genera of variable oxygen requirement and unclassifiable. Comparisons of centered-log ratio (CLR) transformed relative abundances for the three main categories of bacteria (obligate anaerobes, aerobes and facultative anaerobes) by follow-up interval (baseline, middle and late). Data in boxplots (B) are represented as individual values of untransformed relative abundances with median values and interquartile range depicted by the boxplots with comparisons between intervals by non-parametric Wilcoxon tests, with p-values adjusted for multiple comparisons by the Bonferroni method. Bottom Panels (C-D): Relative abundance barplots for oral, lung and gut (F) samples with classification of bacterial genera by plausible pathogenicity into oral commensals, recognized respiratory pathogens and “other” category. Comparisons of CLR-transformed relative abundances for these categories of bacteria by follow-up interval (baseline, middle and late) in boxplots (D), with p-values adjusted for multiple comparisons.”

We also edited the corresponding section in the main results:

“In both oral and lung communities, we found a progressive decline in the CLR-transformed abundance of obligate anaerobes over time”.

We also want to highlight that we performed longitudinal analyses with mixed linear regression models (per the suggestion of Reviewer 2, see R2C7 regarding joint modeling) and regressed CLR-transformed abundance of anaerobes or pathogens in models with random patient intercepts. We added the following text in the Results section as well as a new Figure 4 to illustrate the results of this analysis (see our response to R2C7 for more details).

“Beyond these cross-sectional comparisons of dysbiosis features, we sought to understand the impact of longitudinal changes on patient survival. To that end, we employed joint modeling, a powerful approach that combines longitudinal and survival analysis models. We examined longitudinal quantitative exposures of dysbiosis features (Shannon index, bacterial load, anaerobe and pathogen abundance - details in Methods) in mixed linear regression models in each compartment, and then assessed the impact of longitudinal changes on 60-day survival. Mixed linear regression models demonstrated a progressive decline of Shannon index in all three compartments, with progressive depletion of anaerobes and enrichment for pathogens in the oral and lung compartments (Figure 4). By Cox proportional hazards models adjusted for age, baseline Shannon index and anaerobe or pathogen abundance in the oral and lung compartments were significantly associated with 60-day survival. Integration of longitudinal and survival analyses with joint modeling showed borderline significant effects for pathogen abundance in the oral compartment and anaerobe abundance in the lung compartment.”

R1C14: Line 496: “We processed derived 16S sequences with a custom Mothur-based pipeline and performed analyses at genus level.” Details of the custom pipeline should be given by describing the bioinformatics workflow and/or making available the pipeline code.

Response: We now provide further details of the analytical approach in the Methods.

Code for the custom Mothur pipeline is available here:

https://github.com/MedicineAndTheMicrobiome/AnalysisTools/tree/master/16S_Clust_Gen_Pipeline

Code for the statistical analyses for this project is provided here

<https://github.com/MicrobiomeALIR/MultiCompartmentMicrobiome>

“We processed derived 16S sequences with a custom Mothur-based pipeline (v1.44.1) as previously described (available at

https://github.com/MedicineAndTheMicrobiome/AnalysisTools/tree/master/16S_Clust_Gen_Pipeline). In brief, we deconvoluted sequences from the Illumina MiSeq and processed them through an in-house sequence quality control pipeline, which includes dust low complexity filtering, quality value (QV<25) trimming, and trimming of primers used for 16S rRNA gene amplification, and minimum read length filtering. Trimmed reads shorter than 75bp or those with less than 95% of the bases above a QV of 25 were discarded. Forward and reversed paired reads were merged into contigs, and processed for 16S rRNA gene sequence clustering and annotation pipeline. Sequence taxonomic classifications were performed with the Ribosomal Database Project’s (RDP) naïve Bayesian classifier with the SILVA 16S rRNA database (v138).”

R1C15: Line 570: “To develop these models in each compartment (oral, lung and gut), we used probabilistic graphical modeling (PGM) by considering the 50 most abundant taxa in each compartment along with the Shannon Index.” Please, indicate explicitly what type of abundance was used (absolute, relative, clr, ...) in the MLR model equation.

Response: We utilized clr transformed abundances in the MLR equation. We clarify that in the revised Methods section.

“To develop these models in each compartment (oral, lung and gut), we used probabilistic graphical modeling (PGM) by considering the 50 most abundant taxa (expressed by CLR-transformed relative abundance) in each compartment along with the Shannon Index”.

R1C16: Line 662: PRJNA726955 is not available on NCBI. Please check.

Response: We apologize for any confusion. The Bioproject PRJNA726955 is available on NCBI as shown in the screenshot below.

Internal transcribed spacer (ITS) sequencing for Acute Lung Injury Registry Accession: PRJNA726955 ID: 726955

In critically ill patients, variation in the host inflammatory responses have been associated with poor outcomes. However, the biological mechanisms underlying this heterogeneity have not yet been defined. We investigated whether variation fungal communities (mycobiome) in the respiratory tract are associated with host inflammation and innate immune system activation and clinical and patient centered outcomes.

We collected endotracheal aspirates (ETA), oral wash and plasma samples from critically ill patients. We used extracted DNA from ETAs and performed fungal rRNA gene sequencing (internal transcribed spacer [ITS]) on the Illumina MiSeq platform to characterize the lower respiratory tract mycobiome. We derived diversity metrics to classify patients into strata based on the alpha diversity of the mycobiome and examined for associations with diversity strata and clinical and patient centered outcomes. Less...

Accession	PRJNA726955
Data Type	Raw sequence reads
Scope	Multispecies
Submission	Registration date: 3-May-2021 University of Pittsburgh Medical Center
Relevance	Medical

Project Data:

Resource Name	Number of Links
BioSample	679

No public data is linked to this project. Any recently released data that cites this project will be linked to it within a few days.

R1C17: Line 666: “Primary code and de-identified data for replication of analyses will be available on the github repository (<https://github.com/MicrobiomeALIR/MultiCompartmentMicrobiome>) upon acceptable of the manuscript for Publication.” A confidential link to the reviewer for review purposes would be greatly appreciated and would speed up the review process.

Response: We provide link to our code here:

<https://github.com/MicrobiomeALIR/MultiCompartmentMicrobiome>

R1C18: On a final note, it would be quite helpful if the authors precisely define which genera are included in the classification of bacterial genera into true respiratory pathogens, or oral commensals, or “other” category. For instance, what is the biological meaning of having plausible respiratory pathogens expanded in the gut at the baseline? Who were those? Likewise, how the authors interpret the abundance of the gut commensal *C. albicans* in the lung? Is it a respiratory pathogen? If so, how? Obligate anaerobes progressively declined in both oral and lung communities but not in the gut. Please, indicate here some of the relevant genera (among obligate anaerobes, facultative anaerobes, aerobes and microaerophiles), it will facilitate the comprehension. Likewise, specify the anaerobe abundance in patients with COPD and what the increased pathogen abundance refers to in all three compartments. The fact that receipt of anaerobic spectrum antibiotics was associated with a progressive decrease in obligate anaerobe abundance, without significant effects on pathogen abundance, does it mean that facultative bacteria were spared? How do you explain the the gut-origin taxa enrichment in the lung samples without overt oropharyngeal colonization with such taxa?

Response: We provide further details on classifications of the genera assigned to oxygen requirements and plausible pathogenicity categories, with new supplemental tables. In a new Figure S4, we provide the top 15 abundant taxa in each compartment with examples of the top abundant obligate anaerobes and respiratory pathogens (categories for which most analyses were conducted) to help facilitate comprehension of these categories, as per the Reviewer’s suggestion.

“Figure S3: Top 15 representative taxa in each body compartment (oral, lung and gut). Each taxon is shown by barplots representing the mean relative abundance with the associated standard errors. Members of these top taxa classified as obligate anaerobes (middle panels) or as respiratory pathogens (right panels). The bottom panel displays the longitudinal measurement of bacterial load by 16S qPCR in each compartment.”

With regards to the insightful questions posed by the Reviewer regarding interpretation of our findings, we added the following text in the Discussion section:

“However, we did observe a small subset of patients who had enrichment for gut-origin commensal or pathogenic organisms in their LRT. Such enrichment could not be fully accounted for by URT colonization with similar taxa. These patients with gut-origin bacteria enrichment in their lungs (8.3%) had much worse survival

than the rest of the cohort. This subset of patients may have experienced gut-to-lung bacterial translocation. To further investigate the possibility of gut-to-lung translocation, it would be beneficial to have wider availability of BAL samples to investigate the alveolar spaces more closely. Our non-invasive ETA samples showed that such translocation, if present, affects a small subset of patients at least within the first week of IMV. Therefore, efforts focused on preventing lung dysbiosis and pathogen colonization will need to consider primarily the URT-to-LRT ecosystem and secondarily, the possibility of gut-to-lung translocation.”

“Integration of fungal sequencing data further enhanced our view of the microbial communities, revealing patients who had enrichment for C.albicans and experienced worse outcome. Presence of C.albicans in the LRT may not signify clinical pneumonia by conventional criteria, yet may represent a state of dysbiosis with potential adverse effects from C.albicans on host epithelial integrity and immune response.”

We provide detailed new Tables S2 and S3 with our operational classifications of genera by oxygen requirements and plausible pathogenicity.

Reviewer #2 (Remarks to the Author):

Kitsios et al. present a longitudinal multicenter cohort study investigating dynamic changes in the oral, lung, and gut microbiota in hospitalized critically ill patients and the impact of these changes on pertinent clinical outcomes. This is an important study that further advances our understanding of the prognostic relevance of microbiota signatures in critically ill patients.

The investigators astutely point out that most microbiota studies involving critically ill patients are limited by statistical models that incorporate only a single microbial signature at a fixed sampling timepoint.

They found that dysbiotic signatures emerged in all three body compartments (lung, gut, and oral cavity), predominantly characterized by a decrease in alpha diversity, a depletion of obligate anaerobe bacteria, and increased enrichment in more traditionally pathogenic taxa. In addition, they used DMM and bacterial-fungal SNF clustering methods to identify microbial signatures in the lungs (e.g., low diversity in the case of DMM clusters, high pathogen abundance in the case of bacterial-fungal SNF clusters) that were associated with worse survival.

Overall, this is a well-conducted study that addresses several novel questions. Strengths of this study include its large patient sample size (with a vast number of oral, LRT, and gut samples), its use of both derivation and validation cohorts, its use of longitudinally collected samples, and its use of methods that extend beyond 16S rRNA gene sequencing (which is limited solely to bacterial taxonomy). The manuscript contains a substantial amount of data but is concisely and clearly written.

Response: Thank you for your very balanced, positive and detailed assessment of our manuscript.

Major comments:

R2C1. The derivation cohort was more than 90% white/Caucasian, which may limit generalizability, although their findings were validated in more racially diverse cohorts (72-76% white). Nonetheless, this should be highlighted as a limitation.

Response: We agree with this comment, and added the following limitation in the Discussion section:

“Our derivation cohort had limited racial/ethnic diversity consistent with the demographics of the catchment populations for our ICUs; therefore our results require independent validation in more diverse patient populations.”

R2C2. I was surprised to see a lack of taxonomic differences between the oral and lung microbiota

(Figure 1H). To clarify, was this looking collectively at all oral and LRT samples? Is it possible that differences might emerge if you stratify by sample collection timepoint? Do the investigators think there would have been differences if they were comparing oral samples to BAL samples rather than to tracheal aspirates?

Response: We used a conservative threshold of $\log_{2}FC \geq 1.5$ in our taxonomic composition comparisons. There were multiple taxa with statistically significant differences (adjusted $p < 0.05$, $n = 90$) that did not exceed this stringent threshold, as we wanted to focus on abundant taxa with large effect sizes. We did not detect more significant deviation of oral-lung microbiota by sample collection timepoint. We added the following statement in the Discussion:

“However, we may have missed important microbiota variability closer to the alveolar space, including potentially stronger deviation from URT microbiota, higher signal of gut-to-lung microbiota translocation, as well as better delineation of longitudinal host-response biomarkers in BAL fluid.”

R2C3. Did the authors perform any type of contamination analysis? They very clearly demonstrate low biomass in their control samples. However, there is no designation of which individual ASVs/genera potentially represent contaminants. This is due mainly to the fact that microbes were clustered according to their oxygen requirement and pathogenicity (two classification schemes with clinical relevance in one prior study of patients with aspiration pneumonia) rather than genera.

Response: We have included a new panel in Figure S1 that illustrates the most commonly detected taxa in negative control samples, indicative of contamination. We agree with the Reviewer that negative controls had a reassuringly low signal of bacterial DNA detection. We did not perform any further filtering for taxa detected in negative controls.

“D. Top taxa detected in negative control samples, indicating possible contamination. Overall, such taxa were detected at very low levels (mean number of reads <100).”

We added the following text in the Methods:

“Examination of taxonomic composition of negative control samples revealed very low numbers of reads for commonly detected taxa (mean <100 reads, Figure S1) and we did not filter any taxa from clinical samples.”

R2C4. Can the authors provide a figure demonstrating what happens to bacterial load in all three compartments over time?

Response: We added a panel in new Figure S4 with these data.

“Figure S4: Top 15 representative taxa in each body compartment (oral, lung and gut). Each taxon is shown by barplots representing the mean relative abundance with the associated standard errors. Members of these top taxa classified as obligate anaerobes (middle panels) or as respiratory pathogens (right panels). The bottom panel displays the longitudinal measurement of bacterial load by 16S qPCR in each compartment.”

We have also conducted longitudinal modeling of bacterial load by 16S qPCR and provide these results in a new Figure 4 (as part of the joint modeling analysis).

Additionally, we conducted a new analysis with mixed linear regression models to examine the impact of antibiotics and steroids on longitudinal bacterial load in the three compartments (as quantified by 16S qPCR). We found significant effects of different antibiotic scores for the oral and gut bacterial load. We report these results in the revised Table S4.

“Table S4: Mixed linear regression models for the examination of the effects of antibiotics and steroids on features of dysbiosis in samples from all three compartments. We examined the antibiotic exposure coded in three different ways: i) anaerobic coverage, ii) a numerical scale that included duration, timing and type, and iii) the Narrow Antibiotic Treatment (NAT) score. Each effect was adjusted for the study day from enrollment. The p-values of the mixed effects models with random patient intercepts are shown for each endpoint (columns) and significant values are highlighted in bold.”

Variables	Shannon index	Bacterial load (16S qPCR)	Obligate anaerobe abundance	Respiratory Pathogen abundance
Oral				
Anaerobic_spectrum	0.422	0.054	0.023	0.526
Antibiotic_score	0.535	0.010	0.857	0.607
NAT_score	0.248	0.006	0.262	0.075
Steroids_score	0.499	0.793	0.542	0.116
Lung				
Anaerobic_spectrum	0.645	0.147	0.037	0.756
Antibiotic_score	0.605	0.841	0.578	0.842
NAT_score	0.456	0.064	0.734	0.262
Steroids_score	0.862	0.593	0.400	0.782
Gut				
Anaerobic_spectrum	0.348	0.801	0.002	0.016
Antibiotic_score	0.067	0.006	0.004	0.031
NAT_score	0.590	0.883	0.008	0.040
Steroids_score	0.671	0.437	0.083	0.218

R2C5. Was there any consideration given to presenting these data with stratification according to genera rather than oxygen requirement and pathogenicity designations? What was the basis for these groupings? They only provide a single citation justifying this approach.

Response: We provide further justification for our operational classifications of bacterial taxa by oxygen requirements and plausible pathogenicity. We provide citations to multiple related papers supporting the biological and clinical relevance of this classification scheme.

“We then examined the discovered bacterial taxa at genus level and classified them by two different classification schemes with clinical relevance. First, we considered the oxygen requirements of each bacterial taxon, given the relevance of oxygen metabolism in critically ill patients on invasive mechanical ventilation who receive variable amounts of inspired oxygen. Recent research has investigated the impact of hyperoxia in LRT microbiota, as well as the association between anaerobic spectrum antibiotics with anaerobe bacteria abundance in the respiratory and gastrointestinal tract.”

“Next, we classified organisms based on their plausible pathogenicity. Prior research has examined the associations between oral-origin bacteria in the LRT (i.e. lung commensals) with innate immunity and clinical outcome, as well as the importance of detecting highly abundant typical respiratory pathogens as causal factors of LRT infection in critically ill patients. We therefore utilized operational definitions of plausible pathogenicity of detected taxa as follows.”

“We then sought to understand the longitudinal composition of microbial communities when classified into clinically relevant categories of bacterial taxa. Recent evidence implicates loss of commensal anaerobic bacteria from the gastrointestinal or respiratory tract with adverse outcome in critical illness. Therefore, we classified bacteria in terms of their oxygen requirements (obligate anaerobes, facultative anaerobes, aerobes, microaerophiles, variable or unclassifiable, details in Supplemental Tables). Additionally, we classified bacteria by plausible respiratory pathogenicity (oral commensals, recognized respiratory pathogens or other) due to their direct implications in prevalent or incident pneumonia in the ICU.”

We elected to examine for global features of microbial communities rather than individual taxa abundance, because we wanted to derive generalizable and robust signatures of microbiota.

We provide the classifications in new Tables S2 and S3. We used extensive literature searches to derive classifications for each genus by 16S, but recognize that these classifications are operational and subject to

implicit assumptions about specific genera. We consider this framework reproducible, as our results can be subjected to sensitivity analyses for revised classification based on emergent evidence.

R2C6. What were the major taxonomic differences between DMM and weighted SNF clusters? This should be more clearly stated.

Response: We now show in graphical form (heatmaps) the major differences between the DMM clusters.

“Figure S7: Heatmaps of relative abundance for the top 15 taxa in each compartment grouped by bacterial DMM clusters.”

We elected to remove the SNF clusters due to the large amount of data and analyses added in the Revised manuscript, which would make interpretation of the SNF cluster results more challenging.

R2C7. In Figure 4, are the predictors used in the survival analyses DMM and SNF designations at any collection timepoint? Or does this only constitute the first sampling timepoint? If it is any sampling timepoint, the study loses its ability in part to look at how dynamic changes affect survival, which is really what drives home the novelty of this investigation. They point out that there was very little shift between DMM clusters based on the timing of sample collection. Then again, they also point out how the number of samples dwindled in later sample collection timepoints. Nonetheless, these analyses could be modeled using joint modeling of longitudinal and time-to-event data (see Henderson et al., Biostatistics 2000) with reporting of association strength estimates.

Response: Thank you for this very insightful comment. We have implemented joint modeling analyses per your suggestion, which offered further insights into our dataset.

We first clarify that the survival analyses with the cluster assignments represent baseline cluster assignments. We consider such findings important as a baseline assessment of microbiota profiles can offer prognostication for patient-centered outcomes, independent of other clinical predictors. For prognostic purposes, a single baseline microbiome measurement is more practical and generalizable than longitudinal measurements. Motivated by your suggestion, we also performed a survival analysis by longitudinal classifications to lung DMM clusters, and added the following text in the Results.

“Furthermore, by examining the longitudinal evolution of lung DMM clusters, patients who remained in the low diversity cluster from the baseline to the middle interval (“Low Diversity Persisters”, Figure 4) had significantly

worse survival than other patients with available follow-up samples (age-adjusted HR=2.73 [1.19-6.42], p=0.02).”

We concur with the Reviewer that insights into the dynamic evolution of the microbial communities and their impact on patient outcomes had been limited to date, and our dataset can answer some of these questions.

We implemented joint modeling (mixed linear regression with random patient intercepts combined with cox-proportional hazards model for 60-day survival, adjusted for age) with the *joiner* package in R. We constructed such models for the exposures of longitudinal change of Shannon index (rarefied), bacterial load (by 16S qPCR), Anaerobe abundance (CLR-transformed) and Pathogen abundance (CLR-transformed), separately in each body compartment (oral, lung and gut). Then, in graphical format we provide i) the beta-coefficients with 95% confidence intervals for the longitudinal change of these microbiota variables during follow-up, ii) the age-adjusted hazards ratios with 95% confidence intervals from Cox-proportional hazards models for the baseline values of each microbiota variable (Shannon, bacterial load, anaerobe and pathogen abundance) on 60-day survival, and iii) the joint-modeling adjusted beta-coefficient for the effect of each variable on survival. Importantly, we show that Shannon Index, Anaerobe and Pathogen Abundance in the lung compartments were prognostic as baseline variables, but their longitudinal change were not (with the exception of a borderline significant result for anaerobe abundance by joint modeling).

We consider these results particularly important, and express our gratitude to the Reviewer for this very constructive criticisms. In that regard, we now present a new main Figure with the joint modeling analyses, and incorporate in that same Figure the previous survival analysis by DMM clusters.

We added a new Figure 4:

“Figure 4: Lung dysbiosis features and clusters predict 60-day survival. A-B: Forest plots of effect sizes (point estimates and 95% confidence intervals) for dysbiosis features (Shannon index, bacterial load, anaerobe and pathogen abundance) in three different models: i) mixed linear regression models with random patient intercepts for the longitudinal change of dysbiosis features during follow-up sampling, ii) the age-adjusted hazards ratios from Cox-proportional hazards models for the baseline values of each feature on 60-day survival, and iii) joint-modeling with adjusted beta-coefficient for the effect of each longitudinally-measured feature on survival. Joint modeling showed that pathogen abundance in the oral compartment and anaerobe abundance in the lung compartment had borderline statistically significant effects on 60-day survival. Joint-models for bacterial load by qPCR did not converge due to low number of longitudinal measurements. C. Kaplan-Meier curves for 60-day survival from intubation stratified by oral (A), lung (B) and gut (C) bacterial DMM clusters. The Low-Diversity lung DMM cluster was independently predictive of worse survival (adjusted Hazard Ratio = 2.22 (1.0.7-4.63), p=0.03), following adjustment for age, sex, history of COPD, immunosuppression, severity of illness by sequential organ failure assessment (SOFA) scores and host-response subphenotypes. Longitudinal analysis of lung DMM clusters showed that patients who remained in the low diversity cluster from the baseline to the middle interval (“Low Diversity Persisters”) had significantly worse survival than other patients with available follow-up samples (age-adjusted HR=2.73 [1.19-6.42], p=0.02).”

A. Oral microbiota features and survival

B. Lung microbiota features and survival

C. Survival analyses by bacterial DMM clusters

We added the following writing in the Methods section:

“In each body compartment (oral, lung, and gut), we examined for the impact of dynamic changes in microbiota features (rarefied Shannon Index, bacterial load by 16S qPCR, CLR-transformed Anaerobe abundance and CLR-transformed Pathogen abundance) on 60-day survival using joint modeling. The joint models combined a mixed linear regression model with random patient intercepts for measuring the longitudinal changes of each feature during sampling follow-up, and a Cox-proportional hazards model adjusted for age. Joint modeling offers the advantage of providing estimates of time-related associations with outcome and can handle informative censoring, which may have impacted our follow-up sample availability (e.g. in the case of patients with early mortality or patients with rapid clinical improvement and discharge from the ICU). We built joint models with the joineR package, and reported in graphical format: i) the beta-coefficients with 95% confidence intervals (CI, estimated via bootstrapping at 100 iterations) for the longitudinal change of the microbiota variables during follow-up from the mixed linear regression models, ii) the age-adjusted hazards ratios with 95% confidence intervals from Cox-proportional hazards models for the baseline values of each microbiota variable on 60-day survival, and iii) the joint-modeling adjusted beta-coefficient for the effect of each variable on survival.”

In the Results section, we added the following text:

“Beyond these cross-sectional comparisons of dysbiosis features, we sought to understand the impact of longitudinal changes on patient survival. To that end, we employed joint modeling, a powerful approach that combines longitudinal and survival analysis models. We examined longitudinal quantitative exposures of dysbiosis features (Shannon index, bacterial load, anaerobe and pathogen abundance - details in Methods) in mixed linear regression models in each compartment, and then assessed the impact of longitudinal changes on 60-day survival. Mixed linear regression models demonstrated a progressive decline of Shannon index in all three compartments, with progressive depletion of anaerobes and enrichment for pathogens in the oral and lung compartments (Figure 4). By Cox proportional hazards models adjusted for age, baseline Shannon index and anaerobe or pathogen abundance in the oral and lung compartments were significantly associated with

60-day survival. Integration of longitudinal and survival analyses with joint modeling showed borderline significant effects for pathogen abundance in the oral compartment and anaerobe abundance in the lung compartment. Nonetheless, baseline microbiota features had stronger effect sizes associated with survival.”

In the Discussion section, we added the following text:

“Notably, we modeled the longitudinal change in dysbiosis features and its impact on survival with joint modeling. Joint modeling is a flexible approach that can mitigate some of the effects of informative censoring. The latter is particularly relevant for translational studies in the ICU, because patients who experienced early mortality or those who improved quickly and were discharged from the ICU could not contribute later follow-up samples. Our joint models revealed that baseline features in the lung compartment (Shannon index and anaerobe or pathogen abundance) were predictive of survival, whereas their longitudinal changes were not, except for marginal effects of anaerobe abundance. Such results may indicate that the communities formed by host-microbiota interactions early post-intubation are already representative of a LRT infection or dysbiosis state. Therefore, subsequent changes among those who remain intubated in the ICU may be less consequential for the overall outcome compared to their starting state. Nonetheless, longitudinal observations were limited by lower number of observations, which has likely limited the statistical power of joint models, as indicated by the wide CI in effect estimates.”

R2C8. The authors clearly state that use of endotracheal aspirate samples to represent the lung microbiota as opposed to reference standard BAL is a limitation of their study. They should comment more on how their findings might have differed if they used BAL samples.

Response: We recognize this limitation and added the following text in the Discussion:

“Recent research has shown the ability to derive robust microbiota signatures from ETA samples in patients on IMV.⁵¹ However, we may have missed important microbiota variability closer to the alveolar space, including potentially stronger deviation from URT microbiota, higher signal of gut-to-lung microbiota translocation, as well as better delineation of longitudinal host-response biomarkers in BAL fluid.¹⁸ In a limited comparison of two subjects with synchronous ETA-BAL sampling, we found that in a case of *Achromobacter xylosoxidans* pneumonia, both ETA and BAL sample showed community dominance (>90% relative abundance) by *Achromobacter* genera, whereas in a case of culture-negative pneumonia diagnosis, taxonomic concordance between ETA and BAL sample was more limited. These results are consistent with a previous comparison of ETA vs. mini-BAL metagenomics, in which case higher taxonomic concordance was seen for cases with culture-positive pneumonia. Thus, the reliability of ETA biospecimens for profiling airspace microbiota may be context dependent, and further research is needed with BAL biospecimens when available.”

We also examined for taxonomic similarity between ETA and BAL samples in a limited comparison of 2 subjects that had available samples of each type obtained at the same time. We added a new Figure S2 that illustrates this comparison. Due to small sample size, we did not perform any formal statistical testing but made observations on the dominant taxa by each sample type.

“**Figure S2: Taxonomic comparison of top 10 abundant taxa between Endotracheal Aspirate (ETA) and Bronchoalveolar Lavage (BAL) samples that were synchronously obtained from the same subject.** For subject A, both ETA and BAL samples showed near complete community dominance by *Achromobacter* genera (>90%). Subject A was clinically diagnosed with *Achromobacter xylosoxidans* pneumonia based on BAL microbiologic cultures, consistent with 16S-Seq data by both ETA and BAL. For subject B that had higher alpha diversity than subject A, there was limited taxonomic concordance between ETA and BAL sample, with several different top abundant taxa between sample types (top abundant taxon *Veillonella* in the ETA sample vs. *Gemmataceae* taxa in the BAL sample). Subject B had no bacterial growth in clinical BAL cultures and was diagnosed as culture-negative pneumonia, a diagnosis that was not supported by 16S-Seq of ETA or BAL sample.”

We added the following text in the Results section:

“In a limited comparison of two subjects with available synchronous ETA and BAL samples, high compositional concordance was shown for one subject in whom LRT community dominance by *Achromobacter* was shown for both ETA and BAL analysis, whereas for the other subject, taxonomic overlap between ETA and BAL was more limited (Figure S2).”

R2C9. The authors point out a lack of association between inflammatory sub-phenotypes and DMM microbiota clusters. Regarding the lungs, multiple studies to date have shown that the lower airway microbiota affects lower airway immune tone in both diseased and healthy individuals. Do they investigators have any insight as to why no association was noted in their cohort? Is this because of their use of tracheal aspirates rather than BAL?

Response: Thank you for this opportunity to clarify our findings and improve presentation of our results. We recognize that due to the bulk of our analyses, many results have been only shown in the Supplementary material and in rather low resolution, therefore limiting the ability to be interpreted and synthesized appropriately. We aimed to improve the presentation of our display items in this revision and use more and larger supplemental figures.

We would like to clarify that we did find significant associations between LRT (ETA) microbiota and host-biomarkers, both in the UPMC-ARF and UPMC-COVID cohorts, consistent with our hypotheses and prior literature.

We added the following text in the Results:

“We found several significant correlations (Figure S9A-C), with typical pathogens correlating with ETA or plasma inflammatory biomarkers, such *Klebsiella* or *Staphylococcus* genera positively correlating with ETA fractalkine and Ang-2 levels, whereas *Escherichia-Shigella* abundance correlating with plasma TNFR1 and IL-6 levels. Conversely, typical oral commensals (e.g. *Rothia*, *Streptococcus*, *Prevotella* etc.) were inversely correlated with plasma sTNFR1 or sRAGE.”

“Patients assigned to the low diversity cluster at baseline had higher ETA levels of sTNFR1, as well as higher plasma Ang-2 compared to the high diversity cluster ($p < 0.05$, Figure 5B). By individual taxa abundance, typical pathogen abundance was correlated with intensified ETA inflammation (e.g. *Klebsiella* correlated with higher levels of ETA sTNFR1 and IL-6), several oral commensals were correlated with higher ETA levels of sRAGE (such as *Streptococcus*, *Rothia* and *Veillonella*) potentially indicating higher degree of lung epithelial injury, whereas *Prevotella* abundance was inversely correlated with plasma levels of inflammatory and tissue injury biomarkers.”

We now provide two new Figures for microbiota-biomarker associations in the UPMC-ARF cohort, and one new Figure for better illustration of microbiota-biomarker relationships in the UPMC-COVID cohort.

“Figure S9. Microbiota correlate with host response biomarkers at both the lung compartment and at a systemic level. A-C: Heatmaps of correlations between the top 20 abundant taxa in the oral (A), lung (B) and gut (C) compartment with 10 host response biomarkers measured in plasma samples (top 10 rows) and endotracheal aspirate (ETA) supernatant samples (bottom 10 rows) in each heatmap. ETA biomarker values were adjusted for total protein concentration in each sample. Statistically significant correlations adjusted for multiple testing (Benjamini-Hochberg method) are shown with crosses (“+”) and the direction of the correlation is color coded. D: Comparisons of ETA and plasma biomarkers between bacterial DMM clusters. The low diversity bacterial DMM cluster (brown) had significantly higher levels of plasma sTNFR1, sRAGE and procalcitonin levels.”

“Figure S12: Lung bacteria correlate with host response biomarkers at both the lung compartment and at a systemic level in patients with severe COVID-19. Heatmap of correlations between the top 20 abundant taxa in lung samples with 10 host response biomarkers measured in plasma samples (top 10 rows) and endotracheal aspirate (ETA) supernatant samples (bottom 10 rows) in each heatmap. ETA biomarker values were adjusted for total protein concentration in each sample. Statistically significant correlations adjusted for multiple testing (Benjamini-Hochberg method) are shown with crosses (“+”) and the direction of the correlation is color coded.”

We recognize the caveat though that we had fewer datapoints with ETA than plasma biomarkers, and therefore the taxa-ETA biomarker correlations had smaller effective sample sizes.

Minor comments:

R2C10. A number of abbreviations are used in Figure S1 that are not clearly or accessibly defined.
 Response: All abbreviations are now clearly defined.

R2C11. As part of their supplement, the authors should specify which specific taxa/genera were placed into the oxygen requirement and pathogenicity categories.
 Response: We added extensive tables with all taxa classifications (Tables S2 and S3)

R2C12. The authors might consider regressing bacterial burden against the clinical covariates studied in other linear regression models.
 Response: We added these models in the new Figure 4, new Figure S6, as well the mixed linear regression models in Table S4.

See for example the new Figure S6 that includes bacterial load as dependent variable in linear regression models.

“Figure S6: Clinical variables associated with alpha diversity (Shannon Index), bacterial load (by 16S qPCR), obligate anaerobe and respiratory pathogen abundance in baseline samples from the three body compartments. Clinical variables are shown on the y-axis, and R² of linear regression models for Shannon index (obtained post rarefaction), bacterial load, anaerobe abundance (CLR-transformed), and pathogen abundance (CLR-transformed) are shown on the x-axis. Statistically significant associations (p<0.05) are shown with large bubbles and direction of association is color coded.”

R2C13. How was “immunosuppression” defined in regression models?

Response: We provide the definition of immunosuppression (categorical variable: yes/no) in the footnote of Table 1.

“Immunosuppression was broadly defined as receipt of chronic steroids, alkylating agents, antimetabolites, biologics, calcineurin inhibitors, mycophenolate, active chemotherapy for cancer, or diagnosis of untreated immunodeficiency.”

R2C14. I might consider adding grid lines to the images in Figure S4 to ease readability.

Response: We added soft grid lines in these bubble plots. This Figure is now Figure S6 and available for review in our response to R2C12 above.

REVIEWERS' COMMENTS

Reviewer #1 (Remarks to the Author):

I think that the authors have adequately addressed the comments made by this reviewer in the revised version of the manuscript. Therefore, I have no further comments.

Reviewer #2 (Remarks to the Author):

The reviewers have thoughtfully responded to all of my comments. Any concerns that I initially raised that could be addressed were done so in a clear and succinct manner. Other concerns that could not be directly addressed are now clearly stated as limitations of their work. Overall, I was impressed by the thought and care the authors put into their revised submission. As stated previously, this is an important study that sheds light into the role of microbial dysbiosis on outcomes in critically ill patients. I find the manuscript acceptable for publication in its current form and have no further concerns.